# Seeing Through the Brain: New Insights from Decoding Visual Stimuli with fMRI

**Zheng Huang**[1][*] **Enpei Zhang**[1][*] **Weikang Qiu**[2]**, Yinghao Cai**[1]**, Carl Yang**[3]**, Elynn Chen**[4]**,**
**Xiang Zhang**[5]**, Rex Ying**[2]**, Dawei Zhou**[6]**, Yujun Yan**[1]

[1]Dartmouth College    [2]Yale University    [3]Emory University    [4]New York University
[5]UNC Charlotte    [6]Virginia Tech

## Abstract

Understanding how the brain encodes visual information is a central challenge in neuroscience and machine learning. A promising approach is to reconstruct visual stimuli—essentially images—from functional Magnetic Resonance Imaging (fMRI) signals. This involves two stages: transforming fMRI signals into a latent space and then using a pre-trained generative model to reconstruct images. The reconstruction quality depends on how similar the latent space is to the structure of neural activity and how well the generative model produces images from that space. Yet, it remains unclear which type of latent space best supports this transformation and how it should be organized to represent visual stimuli effectively.

We present two key findings. First, fMRI signals are more similar to the text space of a language model than to either a vision-based space or a joint text–image space. Second, text representations and the generative model should be adapted to capture the compositional nature of visual stimuli, including objects, their detailed attributes, and relationships. Building on these insights, we propose **PRISM**, a model that **P**rojects f**MRI** s**I**gnals into a **S**tructured text space as an inter**M**ediate representation for visual stimuli reconstruction. It includes an object-centric diffusion module that generates images by composing individual objects to reduce object detection errors, and an attribute/relationship search module that automatically identifies key attributes and relationships that best align with the neural activity. Extensive experiments on real-world datasets demonstrate that our framework outperforms existing methods, achieving up to an $6\%$ reduction in perceptual loss. These results highlight the importance of using structured text as an intermediate space to bridge fMRI signals and image reconstruction. Codes are available at `https://github.com/GraphmindDartmouth/PRISM`.

## 1 Introduction

Decoding visual stimuli from brain activity provides a unique lens into human perception (Naselaris et al., 2011; Haufe et al., 2014). A central approach uses fMRI signals—which measure neural activity through blood-oxygen-level-dependent responses—to reconstruct the images perceived by subjects (Allen et al., 2022; Chang et al., 2019; Luo et al., 2023). Recent advances in deep generative models have significantly improved these reconstructions, deepening our understanding of visual representation in the brain (Chen et al., 2023) and enabling applications in brain-computer interfaces (Sitaram et al., 2008) and brain-driven content generation (Wang et al., 2024a; Qiu et al., 2025).

FMRI-to-Image reconstruction involves two stages: mapping fMRI signals into a latent space and then generating images from that space. Its success depends on the similarity between the latent space and neural activity (alignment) and how well the generative model produces high-quality images. While recent studies (Scotti et al., 2023; 2024; Mai et al., 2024) focus on enhancing image quality using advanced generative models (Podell et al., 2023; Xu et al., 2023), alignment remains underexplored. Prior work often assumes that the latent space should match the modality of the stimuli, i.e., using vision model representations to reconstruct visual stimuli (Scotti et al., 2023; Wang et al., 2024c; Xia et al., 2024). Some studies incorporate auxiliary semantic information from

---

[*]Equal contribution. Correspondence to zheng.huang.gr@dartmouth.edu.

language models (LMs) (Lin et al., 2022; Quan et al., 2024), but still rely on vision-based representations as the core latent space. In contrast, we question whether matching the modality of visual stimuli is truly essential for reconstruction. In addition, prior work suffers from limited reconstruction quality due to a unified hidden representation that conflates objects and their attributes, often causing object detection errors, e.g., generating a tiger instead of a gray, tiger-striped cat (Section A of the Appendix). This reflects a fundamental mismatch with human visual processing, which is object-centric and compositional rather than holistic (Marr, 1980; Bracci & Op de Beeck, 2023). Overcoming this limitation calls for generative models that explicitly capture the compositional structure of human perception.

To address these issues, we propose **PRISM**, a model that **P**rojects f**MRI** s**I**gnals into a **S**tructured text space as an inter**M**ediate representation for image reconstruction. To identify the most effective intermediate space, we compare fMRI signals with representations from pre-trained vision, language, and vision–language models using established metrics (Wang et al., 2020; Murphy et al., 2024; Keskar et al., 2016). Unlike prior work assuming vision-based representations are essential, our first finding (Section 3.1) shows that fMRI signals align more closely with the text space of an LM, motivating the use of solely text as a bridge for reconstruction. Building on this, our second finding reveals that reconstruction quality improves when the text and the generative model are adapted to capture the compositional and relational nature of visual stimuli—encompassing objects, their attributes, and their relationships. Guided by these insights, we develop two core modules: an object-centric diffusion module that adapts the diffusion model to generate images by composing individual objects, and an attribute/relationship search module that uses a vision–language model (VLM) to automatically identify object attributes and relationships aligned with neural activity, providing structured guidance for reconstruction. Our contributions are summarized as follows:

- **Novel Findings:** To our knowledge, we are the first to show that accurate visual stimuli reconstruction can be achieved without image-based latent representations, with LM text space effectively bridging brain activity and generative models. Furthermore, we find that adapting this text space and the generative model to capture the compositional and relational nature of visual images further improves reconstruction quality.
- **Novel Framework:** Motivated by our empirical findings, we introduce a new fMRI-to-image reconstruction framework that adapts diffusion models for object-centric generation and leverages VLMs to automatically identify brain-aligned object attributes and relationships that can optimally guide the reconstruction.
- **Comprehensive Experiments:** Extensive evaluations on real-world fMRI datasets demonstrate that our method achieves up to an $6\%$ reduction in perceptual loss compared to state-of-the-art models, highlighting the effectiveness of our framework.

## 2 PRELIMINARY

**Notations.** In our work, we denote the set of fMRI samples collected during image viewing as $\mathcal{X}$. Each sample is a preprocessed 1D vector $\mathbf{x}_i \in \mathbb{R}^v$, capturing neural activity across $v$ voxels selected from brain regions (Scotti et al., 2023). The dataset is split into training and test subsets, with superscripts indicating the split. For example, we denote the training set with $N$ samples as: $\mathcal{X}^{\text{train}} = \{\mathbf{x}_1, \ldots, \mathbf{x}_N\}$. The corresponding image stimulus for the $i$-th sample is denoted as: $\mathbf{Y}_i \in \mathbb{R}^{H \times W \times 3}$, which contains $m$ objects.

**Problem Setup.** Our goal is to reconstruct the visual images that subjects viewed during fMRI recording. Formally, we seek to learn a reconstruction function $\mathcal{F} : \mathbb{R}^v \rightarrow \mathbb{R}^{H \times W \times 3}$ that maps each fMRI sample $\mathbf{x}_i$ to its corresponding image stimulus $\mathbf{Y}_i$.

**Diffusion Model.** Diffusion models (Rombach et al., 2022; Zhang et al., 2023) are a class of generative models that synthesize data by learning to reverse a multi-step noising process. Starting from Gaussian noise, they iteratively denoise a latent variable over a fixed number of timesteps using a denoising network—typically a U-Net—conditioned on the current timestep $t$. This process gradually produces samples resembling the training distribution. To incorporate external inputs such as text, the denoising network can take a conditioning input $\mathbf{C}$, usually text embeddings from a pre-trained encoder (Zhang et al., 2023). Conditioning is implemented via cross-attention mechanisms within the U-Net architecture (Williams et al., 2023), enabling the integration of textual information. In these layers, the latent representations $\mathbf{H}_t \in \mathbb{R}^{h \times w \times d}$ at time $t$ serve as queries, and $\mathbf{C} \in \mathbb{R}^{d_t \times d'}$

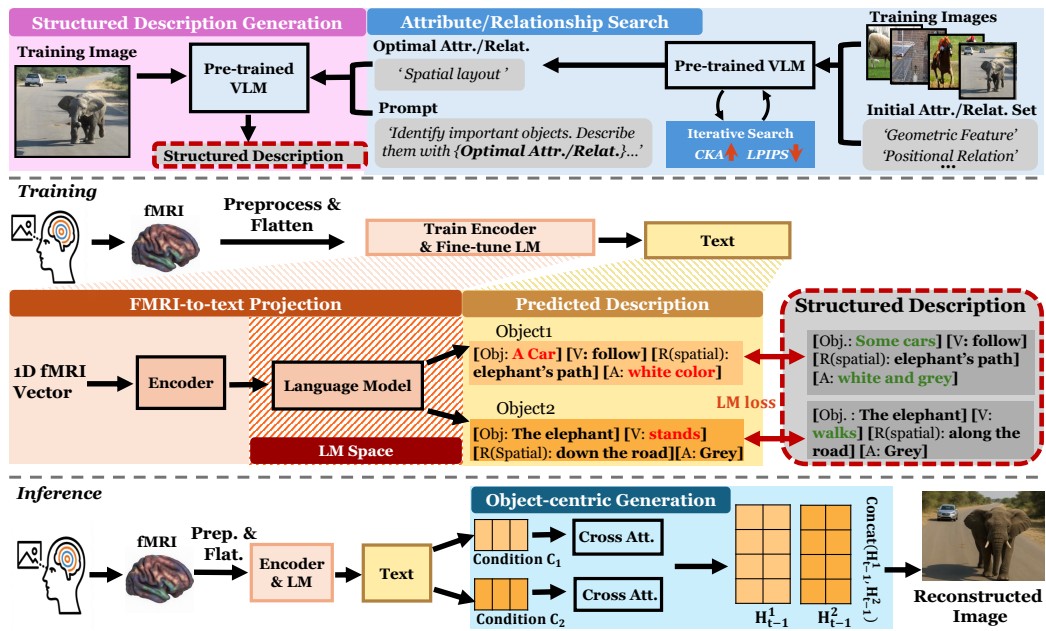

Figure 1: Framework Overview: **PRISM** generates structured text descriptions for each training image using a VLM to iteratively extract brain-aligned object attributes and relationships. These descriptions capture the image's compositional and relational content and serve as supervision to train an encoder and fine-tune a language model to map fMRI signals into the text space. During inference, the model predicts descriptions from fMRI signals, which then guide a pre-trained diffusion model for object-centric image reconstruction.

serves as both keys and values (Williams et al., 2023; Yang et al., 2024):

$$\text{CrossAttention}(\mathbf{H}_t, \mathbf{C}) = \text{softmax}\left(\frac{\phi(\mathbf{H}_t) \cdot \mathbf{W}_Q \cdot (\varphi(\mathbf{C}) \cdot \mathbf{W}_K)^\top}{\sqrt{d_k}}\right) \varphi(\mathbf{C}) \cdot \mathbf{W}_V,$$

where $\mathbf{W}_Q \in \mathbb{R}^{d \times d_k}$, $\mathbf{W}_K \in \mathbb{R}^{d' \times d_k}$, $\mathbf{W}_V \in \mathbb{R}^{d' \times d}$ are projection matrices; and $\phi(\cdot)$ and $\varphi(\cdot)$ are learned transformations. Further details are available in Section B of the Appendix.

## 3 METHOD

In this section, we present our framework, **PRISM**, for fMRI-to-image reconstruction (Figure 1). We first show that fMRI signals align most strongly with the text space of LMs, compared to the hidden spaces of vision or vision–language models, under established metrics (Section 3.1). This finding motivates our choice of using pure text as the latent space. During training (Section 3.2), we annotate each training image with structured text descriptions that are object-centric, compositional, and relational. To generate these descriptions, we introduce an attribute/relationship search module (Section 3.2.1), which learns optimal prompts to guide the VLM in automatically identifying the key attributes and relationships most aligned with both the fMRI signals and images. These structured descriptions are then used to train an encoder and fine-tune the LM, mapping fMRI signals into the LM text space (Section 3.2.2). At inference time (Section 3.3), the predicted structured descriptions guide an adapted diffusion model to generate object-centric images directly from fMRI signals.

### 3.1 TEXT AS THE LATENT SPACE

We question whether using vision representations as the latent space is truly essential for reconstructing visual stimuli. In this section, we investigate the alignment between different model spaces and fMRI signals using various measures.

**Measuring the alignment between model spaces and fMRI signals.** We examine three representation spaces: (1) the text space of language models, (2) the joint text-image space of vision-language models, and (3) the latent space of vision models. For (2) and (3), image embeddings are extracted directly from the respective models. For (1), we use text embeddings from image captions to represent the stimuli. We extract embeddings by feeding either text or images into different models: T5

and LLaMA3 for text embeddings, LDM (Rombach et al., 2022) and ResNet50 (He et al., 2016) for image embeddings, and CLIP for both modalities.

Alignment is assessed using three metrics: Centered Kernel Alignment (CKA) (Murphy et al., 2024), Canonical Correlation Analysis (CCA) (Wang et al., 2020), and Generalization Gap (Keskar et al., 2016). CKA and CCA are widely used to quantify similarity between representation spaces (Kriegeskorte et al., 2008; Wang et al., 2020). Generalization Gap reflects learnability by measuring the train-test loss difference when mapping fMRI signals to a target space using an MLP. Good alignment yields higher CKA and CCA values and a lower Generalization Gap.

Let $\mathbf{X} = \text{Concat}(\mathbf{x}_1, \ldots, \mathbf{x}_N)$ denote concatenated fMRI samples and $\mathbf{K} = \text{Concat}(\mathbf{k}_1, \ldots, \mathbf{k}_N)$ the corresponding latent representations. With $\mathcal{K}(\cdot)$ as a kernel function, the empirical Hilbert-Schmidt Independence Criterion (HSIC) is: $\text{HSIC}(\mathbf{X}, \mathbf{K}) = \frac{1}{(N-1)^2} \text{tr}\left(\mathcal{K}(\mathbf{X}) \cdot \mathcal{K}(\mathbf{K})\right)$, where $\text{tr}(\cdot)$ denotes the trace operator. The CKA is the normalized form: $\text{CKA}(\mathbf{X}, \mathbf{K}) = \frac{\text{HSIC}(\mathbf{X}, \mathbf{K})}{\sqrt{\text{HSIC}(\mathbf{X}, \mathbf{X}) \cdot \text{HSIC}(\mathbf{K}, \mathbf{K})}}$. We adopt a Gaussian radial basis function (RBF) kernel for $\mathcal{K}$ (Alvarez, 2022; Cortes et al., 2012). CCA identifies linear projections $\mathbf{u} = \mathbf{p}_1^\top \mathbf{X}$ and $\mathbf{v} = \mathbf{p}_2^\top \mathbf{K}$ that maximize their correlation. The first mode captures the dominant shared axis (Wang et al., 2020), with the canonical correlation coefficient: $\rho = \text{corr}(\mathbf{u}, \mathbf{v}) = \text{corr}(\mathbf{p}_1^\top \mathbf{X}, \mathbf{p}_2^\top \mathbf{K})$, reflecting the strongest linear alignment between brain activity and the model space.

**FMRI aligns better with the embedding space of language models.** Our results (Table 1) show that the text space of language models aligns best with fMRI data, outperforming both vision–language and vision-only models across all metrics. Surprisingly, vision–language models, despite integrating both modalities, underperform compared to pure language models. We hypothesize that this is because humans focus more on the meaning of an image rather than pixel-level details (Naselaris et al., 2009; Du et al., 2022). Unlike prior work (Scotti et al., 2023; Wang et al., 2024c; Xia et al., 2024; Lin et al., 2022) that primarily relies on vision representations, our findings motivate using pure text as the latent space.

Table 1: Alignment results between model representations and fMRI data, evaluated using CKA, Generalization Gap, and CCA. The best result is highlighted in red. $\uparrow$ denotes higher is better; $\downarrow$ denotes lower is better.

| | CKA $\uparrow$ | Generalization Gap $\downarrow$ | CCA $\uparrow$ |
|---|---|---|---|
| **T5** | 0.5580 | 0.1132 | 0.8344 |
| **Llama3** | 0.5442 | 0.2216 | 0.8022 |
| **Clip text** | 0.5177 | 0.4532 | 0.7599 |
| **Clip img** | 0.3668 | 0.4860 | 0.7573 |
| **LDM** | 0.1957 | 1.2520 | 0.7215 |
| **Resnet50** | 0.1822 | 1.9800 | 0.6746 |

## 3.2 TRAINING OF PRISM

In this section, we describe the training process of **PRISM**, which consists of automatic structured description generation for training images and encoder training.

### 3.2.1 AUTOMATIC DESCRIPTION GENERATION

We design structured text descriptions as supervision for our framework. To capture the compositional and relational nature of human vision, these descriptions should explicitly distinguish between different objects and their relationships. Generating such descriptions with a VLM relies on carefully crafted prompts that specify the desired attributes and relationships, since many of them are not directly reflected in brain activity. To address this issue, we propose a VLM-assisted approach that automatically learns the most relevant attributes and relationships in an image based on the training data, ensuring they are both meaningful and brain-aligned.

We first show how structured descriptions can be generated from a VLM given a learned keyword $a$, and then present our approach for learning the optimal keyword. Given an image $\mathbf{Y}_i$ and a learned keyword $a$, we construct a prompt $\mathcal{P}(a)$ to guide the VLM in describing the most important objects in $\mathbf{Y}_i$ based on $a$. Formally, the VLM receives the image and the prompt as input and outputs a structured description $D_i^a$:

$$D_i^a = \text{VLM}(\mathbf{Y}_i, \mathcal{P}(a)). \tag{1}$$

The structured description is a list of $m$ object-level tuples along with background information:

$$D_i^a = [(o_1 : d_1 : \text{loc}_1), \ (o_2 : d_2 : \text{loc}_2), \ \ldots, \ (o_m : d_m : \text{loc}_m), bg_i]. \tag{2}$$

Each $o_j$ is an object in the image, $d_j$ is its description containing attributes and relationships with other objects conditioned on keyword $a$, and $\text{loc}_j$ denotes its location (selected from a predefined set). The term $bg_i$ represents the background information of image $\mathbf{Y}_i$. To ensure meaningful generation, we further augment each relation description $d_j$ with a structured header encoding its semantic roles, following the PropBank annotation format (Màrquez et al., 2008; He et al., 2017; Ross et al., 2021; Palmer et al., 2005). A case study on reconstruction with object-level description is provided in Section F of the Appendix.

The choice of keyword $a$ strongly influences the attributes and relationships captured in the object descriptions, which in turn affects the quality of mapping fMRI signals to the text space. Ideally, these descriptions should capture the most important information shared between the fMRI signals and the stimulus images. To avoid manually selecting the keyword, we frame its discovery as a prompt optimization problem and introduce our attribute/relationship search module.

Concretely, given a set of training images $\mathcal{Y}^{\text{train}} = \{\mathbf{Y}_1, \ldots, \mathbf{Y}_N\}$ and the corresponding fMRI signals $\mathcal{X}^{\text{train}} = \{\mathbf{x}_1, \ldots, \mathbf{x}_N\}$, we define the following optimization problem to find the optimal $a$ in the prompt ($\mathcal{P}(a)$) for the VLM:

$$\max_a \quad \sum_{i=1}^N \mathcal{S}\left(\mathbf{Y}_i, \text{Diff}(\text{VLM}(\mathbf{Y}_i, \mathcal{P}(a)))\right) \tag{3}$$

$$\text{s.t.} \quad \text{CKA}\left(\mathbf{X}, \mathbf{K}^a\right) > \beta;$$

$$\mathbf{X} = \text{Concat}(\mathbf{x}_1, \ldots, \mathbf{x}_N); \ \mathbf{K}^a = \text{Concat}(\mathbf{k}_1^a, \ldots, \mathbf{k}_N^a);$$

$$\mathbf{k}_i^a = \text{LM}_{\text{ENC}}(\text{VLM}(\mathbf{Y}_i, \mathcal{P}(a))) \text{ for } i = 1, \ldots, N;$$

where $\mathcal{S}(\cdot, \cdot)$ denotes the similarity score between two images (e.g. negative perceptual loss); Diff is a pre-trained diffusion model that generates images from captions produced by the VLM; Concat indicates the concatenation operation across all training samples; and $\text{LM}_{\text{ENC}}$ is a pre-trained language model to encode captions generated by the VLM. The constraint enforces that the CKA similarity between the fMRI data $\mathbf{X}$ and the caption embeddings $\mathbf{K}^a$ generated using keyword $a$ exceeds a threshold $\beta$, ensuring strong alignment between the fMRI and text spaces. The objective ensures that descriptions derived from the optimal keyword support accurate reconstruction.

To optimize the keyword $a$ in Equation (3), we guide the search along semantic links: keywords with similar meanings tend to yield comparable reconstructions, so generating new keywords based on the semantic relationships of top-performing candidates helps uncover more effective prompt expressions. In the search, we utilize an LLM as a keyword generator and iteratively search for improved keywords in a step-by-step manner. We begin by initializing a keywords set $\mathcal{A}$ with a collection of frequently-used attribute and relationship keywords identified in prior works (Johnson et al., 2015; Lu et al., 2016; Krishna et al., 2017). We expand $\mathcal{A}$ through an $\varepsilon$-greedy search strategy: at each search step, the keyword generator proposes new candidate keywords based on either the top-performing keywords in $\mathcal{A}$ with probability $1-\varepsilon$, or randomly selected keywords from $\mathcal{A}$ with probability $\varepsilon$. Only candidates that exceed the similarity threshold are added to $\mathcal{A}$. This balances refinement of effective keywords and exploration of diverse novel keywords. See Section I in the Appendix for the detailed algorithm and search results.

### 3.2.2 ENCODER TRAINING

We design an encoder to map fMRI signals into the latent space of the language model, using structured and object-centric descriptions as supervision. Specifically, each object's information is independently encoded using an MLP. The resulting representations are concatenated and passed to the language model to generate estimated structured descriptions $\hat{D}_i^a$, which can be expressed as:

$$\mathbf{f}_j = \text{MLP}_j(\mathbf{x}_i), \ j = 1, \cdots, m$$
$$\hat{D}_i^a = \text{LM}(\text{MLP}_g(\text{Concat}(\mathbf{f}_1, \ldots, \mathbf{f}_m))), \tag{4}$$

The language model is fine-tuned using a loss over all $m$ object descriptions (Chang et al., 2024; Gunel et al., 2020):

$$\mathcal{L}_{\text{LM}} = -\sum_{j=1}^m \sum_{t'=1}^T \log p(y_{t'} \mid y_{<t'}, \mathbf{f}_j), \tag{5}$$

where $y_{t'}$ denotes the $t'$-th token in the structured description. This training strategy enables fine-grained alignment between fMRI signals and structured textual descriptions. We first train the MLPs

independently for a fixed number of epochs, then jointly fine-tune the language model and MLPs to maximize overall reconstruction performance.

## 3.3 PRISM INFERENCE: OBJECT-CENTRIC IMAGE GENERATION

The inference process of **PRISM** has two steps: (1) generate structured descriptions $\hat{D}_i^a$ from fMRI signals using the trained encoder and language model, and (2) reconstruct the image by composing objects conditioned on these descriptions with a pre-trained diffusion model: $\hat{\mathbf{Y}}_i = \text{Diff}(\hat{D}_i^a)$.

To reflect the brain's compositional understanding of visual scenes, during inference, we adapt a pre-trained diffusion model to perform compositional image generation, inspired by (Yang et al., 2024). Specifically, given an image $\mathbf{Y}_i$ and its predicted structured description $\hat{D}_i^a$ (from Equation (4)), we extract a set of predicted objects $\{o_j\}_{j=1}^m$ and a background description $\hat{bg}_i$. These are combined into a global context prompt $\hat{p}_0$ and embedded into a conditioning matrix $\mathbf{C}_0$. Similarly, each object description $\hat{d}_j$ is embedded into a corresponding conditioning matrix $\mathbf{C}_j$. These conditioning matrices guide the denoising process via cross-attention (Section 2) from time $t$ to 0. At each step, the hidden representation of object $j$ (or the global context when $j = 0$) at time $t-1$ is computed as: $\mathbf{H}_{t-1}^j = \text{CrossAttention}(\mathbf{H}_t, \mathbf{C}_j)$, where $\mathbf{H}_t$ is the hidden representation of the full image at time $t$. We then resize and concatenate the object representations according to their predicted locations $\hat{\text{loc}}_j$:

$$\mathbf{H}_{t-1}^{\text{cat}} = \Psi(\{\mathbf{H}_{t-1}^j, \hat{\text{loc}}_j\}_{j=1}^m), \tag{6}$$

where $\Psi(\cdot)$ denotes the resizing and spatially-aware concatenation operation. Thus, the image generation model encodes each object independently from its description and then spatially concatenates their hidden representations according to the predicted locations.

To ensure smooth region boundaries and seamless fusion between objects and background, we compute a weighted sum of the global context latent and the object latents:

$$\mathbf{H}_{t-1} = \beta \cdot \mathbf{H}_{t-1}^{\text{cat}} + (1 - \beta) \cdot \mathbf{H}_{t-1}^0, \tag{7}$$

where $\beta$ is a hyperparameter that controls the blending ratio. This process is repeated across denoising steps, enabling structured, object-aware generation aligned with the brain's visual understanding.

## 4 EXPERIMENTS

In this section, we conduct extensive experiments to evaluate **PRISM**, guided by the following questions: (**RQ1**) How well does our framework **PRISM** perform on the image reconstruction task? (**RQ2**) How do different choices of latent space influence the reconstruction quality? (**RQ3**) What is the contribution of each component in our framework to the overall reconstruction performance?

## 4.1 EXPERIMENTAL SETUP

We conduct experiments on three datasets: NSD (Allen et al., 2022), BOLD5000 (Chang et al., 2019), and GOD (Horikawa & Kamitani, 2017). Detailed descriptions of the datasets are provided in Section C of the Appendix. Each method is evaluated by comparing the reconstructed images to the ground truth using five metrics: PixCorr and SSIM for pixel- and structure-level similarity (Wang et al., 2004), LPIPS (Zhang et al., 2018) for human perceptual similarity, and CLIP and Inception V3 two-way identification (Scotti et al., 2024) for semantic consistency based on pretrained model representations. We compare our method against the following baselines: Takagi & Nishimoto (Takagi & Nishimoto, 2023) (Takagi for short), Mindvis (Chen et al., 2023), Mindeye (Scotti et al., 2023), MindBridge (Wang et al., 2024b), NeuralDiffuser (Li et al., 2025), and Mindeye2 (Scotti et al., 2024). To ensure a fair comparison, we use the same generative model, Stable Diffusion 2.1 (Pernias et al., 2023; Rombach et al., 2022), for all methods. We additionally present results for **PRISM** and Mindeye2 (ranked second-best) with the newer SDXL backbone (Podell et al., 2023). We incorporate a negative prompt in our model—a textual constraint that guides the diffusion model to avoid generating undesired visual artifacts, such as distorted object shapes or background clutter. For fairness, we apply the same negative prompt when evaluating all baselines, which leads to performance gains in some cases. More details about the baselines and training are provided in Sections D and E of the Appendix.

Table 2: Comparison of our framework with state-of-the-art methods on three datasets. All methods use Stable Diffusion 2.1 as the backbone unless otherwise specified (+SDXL). Results are reported using PixCorr, SSIM, LPIPS, CLIP and Inception V3 metrics. The best result using the same backbone in each column is highlighted in red. ↑ indicates higher is better and ↓ indicates lower is better.

| NSD | PixCorr ↑ | SSIM ↑ | LPIPS ↓ | CLIP ↑ | Inception V3 ↑ |
|---|---|---|---|---|---|
| **PRISM** | $0.3404_{\pm 0.05}$ | $0.4640_{\pm 0.02}$ | $0.5963_{\pm 0.02}$ | $0.9467_{\pm 0.03}$ | $0.9516_{\pm 0.03}$ |
| **Takagi** | $0.2100_{\pm 0.01}$ | $0.3880_{\pm 0.04}$ | $0.7665_{\pm 0.04}$ | $0.8811_{\pm 0.06}$ | $0.9086_{\pm 0.07}$ |
| **Mindvis** | $0.2736_{\pm 0.06}$ | $0.3868_{\pm 0.06}$ | $0.6789_{\pm 0.02}$ | $0.9000_{\pm 0.05}$ | $0.9135_{\pm 0.05}$ |
| **Mindeye1** | $0.3114_{\pm 0.05}$ | $0.3868_{\pm 0.06}$ | $0.6501_{\pm 0.03}$ | $0.9121_{\pm 0.04}$ | $0.9198_{\pm 0.03}$ |
| **MindBridge** | $0.1802_{\pm 0.03}$ | $0.2823_{\pm 0.02}$ | $0.6977_{\pm 0.03}$ | $0.9427_{\pm 0.02}$ | $0.9242_{\pm 0.03}$ |
| **NeuralDiffuser** | $0.3011_{\pm 0.05}$ | $0.3348_{\pm 0.03}$ | $0.6522_{\pm 0.04}$ | $0.9409_{\pm 0.01}$ | $0.9487_{\pm 0.02}$ |
| **Mindeye2** | $0.3160_{\pm 0.04}$ | $0.4447_{\pm 0.02}$ | $0.6338_{\pm 0.04}$ | $0.9201_{\pm 0.03}$ | $0.9308_{\pm 0.03}$ |
| **PRISM+SDXL** | $0.3645_{\pm 0.02}$ | $0.4983_{\pm 0.04}$ | $0.5563_{\pm 0.02}$ | $0.9600_{\pm 0.01}$ | $0.9765_{\pm 0.01}$ |
| **Mde2+SDXL** | $0.3471_{\pm 0.04}$ | $0.4425_{\pm 0.04}$ | $0.6002_{\pm 0.01}$ | $0.9599_{\pm 0.02}$ | $0.9602_{\pm 0.01}$ |
| **BOLD5000** | | | | | |
| **PRISM** | $0.2315_{\pm 0.01}$ | $0.5341_{\pm 0.02}$ | $0.6198_{\pm 0.02}$ | $0.7720_{\pm 0.03}$ | $0.6601_{\pm 0.07}$ |
| **Takagi** | $0.1815_{\pm 0.03}$ | $0.4418_{\pm 0.06}$ | $0.7558_{\pm 0.06}$ | $0.6990_{\pm 0.04}$ | $0.5667_{\pm 0.01}$ |
| **Mindvis** | $0.2122_{\pm 0.05}$ | $0.4944_{\pm 0.04}$ | $0.6463_{\pm 0.05}$ | $0.7720_{\pm 0.04}$ | $0.5701_{\pm 0.08}$ |
| **Mindeye1** | $0.1942_{\pm 0.01}$ | $0.4838_{\pm 0.03}$ | $0.6913_{\pm 0.04}$ | $0.7288_{\pm 0.03}$ | $0.6222_{\pm 0.07}$ |
| **MindBridge** | $0.1522_{\pm 0.04}$ | $0.3005_{\pm 0.01}$ | $0.6535_{\pm 0.03}$ | $0.7431_{\pm 0.01}$ | $0.6200_{\pm 0.04}$ |
| **NeuralDiffuser** | $0.2036_{\pm 0.04}$ | $0.4005_{\pm 0.02}$ | $0.6899_{\pm 0.01}$ | $0.7609_{\pm 0.01}$ | $0.6522_{\pm 0.03}$ |
| **Mindeye2** | $0.2265_{\pm 0.02}$ | $0.5164_{\pm 0.02}$ | $0.6416_{\pm 0.03}$ | $0.7600_{\pm 0.04}$ | $0.6428_{\pm 0.03}$ |
| **PRISM+SDXL** | $0.2442_{\pm 0.02}$ | $0.5600_{\pm 0.03}$ | $0.5909_{\pm 0.04}$ | $0.7881_{\pm 0.04}$ | $0.6881_{\pm 0.02}$ |
| **Mde2+SDXL** | $0.2310_{\pm 0.04}$ | $0.5185_{\pm 0.02}$ | $0.6186_{\pm 0.02}$ | $0.7503_{\pm 0.04}$ | $0.6556_{\pm 0.02}$ |
| **GOD** | | | | | |
| **PRISM** | $0.2571_{\pm 0.01}$ | $0.5200_{\pm 0.02}$ | $0.6213_{\pm 0.01}$ | $0.8567_{\pm 0.05}$ | $0.8428_{\pm 0.06}$ |
| **Takagi** | $0.2322_{\pm 0.05}$ | $0.4944_{\pm 0.04}$ | $0.6463_{\pm 0.05}$ | $0.7232_{\pm 0.01}$ | $0.7556_{\pm 0.02}$ |
| **Mindvis** | $0.1921_{\pm 0.02}$ | $0.4304_{\pm 0.03}$ | $0.697_{\pm 0.04}$ | $0.7162_{\pm 0.03}$ | $0.6119_{\pm 0.03}$ |
| **Mindeye1** | $0.2286_{\pm 0.03}$ | $0.4766_{\pm 0.02}$ | $0.6807_{\pm 0.05}$ | $0.8093_{\pm 0.01}$ | $0.8002_{\pm 0.04}$ |
| **MindBridge** | $0.1898_{\pm 0.04}$ | $0.4227_{\pm 0.02}$ | $0.6960_{\pm 0.04}$ | $0.8219_{\pm 0.01}$ | $0.7960_{\pm 0.01}$ |
| **NeuralDiffuser** | $0.2006_{\pm 0.02}$ | $0.4253_{\pm 0.03}$ | $0.6836_{\pm 0.02}$ | $0.8501_{\pm 0.01}$ | $0.8372_{\pm 0.03}$ |
| **Mindeye2** | $0.2442_{\pm 0.03}$ | $0.4952_{\pm 0.01}$ | $0.6586_{\pm 0.02}$ | $0.8322_{\pm 0.02}$ | $0.8280_{\pm 0.04}$ |
| **PRISM+SDXL** | $0.2669_{\pm 0.01}$ | $0.5537_{\pm 0.01}$ | $0.5989_{\pm 0.03}$ | $0.8727_{\pm 0.03}$ | $0.8820_{\pm 0.04}$ |
| **Mde2+SDXL** | $0.2500_{\pm 0.04}$ | $0.5511_{\pm 0.04}$ | $0.6224_{\pm 0.01}$ | $0.8678_{\pm 0.02}$ | $0.8556_{\pm 0.01}$ |

## 4.2 EFFECTIVENESS OF **PRISM**

We evaluate our fMRI-to-image reconstruction framework on test data and compare it with state-of-the-art methods.

Table 2 summarizes the results, with all metrics and standard deviations averaged across subjects over five runs. Visualizations of reconstructed examples from the test set are shown in Figure 2. As shown in Table 2, **PRISM** outperforms state-of-the-art methods across all datasets and met-

Table 3: Image QA results on reconstructed NSD images. **Mdvs**, **Mde1** and **Mde2** refer to **Mindvis**, **Mindeye1** and **Mindeye2**, respectively. Results are reported as accuracy, with the best highlighted in red.

| | PRISM | Takagi | Mdvs | Mde1 | Mde2 |
|---|---|---|---|---|---|
| **Acc.↑** | 0.6054 | 0.4011 | 0.5037 | 0.5516 | 0.5765 |

rics, with up to a 6% improvement in LPIPS, indicating higher perceptual similarity to the original images. Unlike baselines such as Mindeye2 (Scotti et al., 2024) and Mindeye1 (Scotti et al., 2023), which often ignore key objects, **PRISM** successfully reconstructs all objects, yielding notable gains across all metrics. These results demonstrate the effectiveness of our framework in translating brain activity into accurate, perceptually aligned image reconstructions.

We further evaluate our method on a question answering (QA) task using reconstructed test images from the NSD dataset. For each image, we retrieve a corresponding question–answer pair from the COCO dataset (Lin et al., 2014) and use Qwen2.5 (Bai et al., 2025) to answer the question based on the generated image. QA accuracy is reported in Table 3. Our method achieves an accuracy

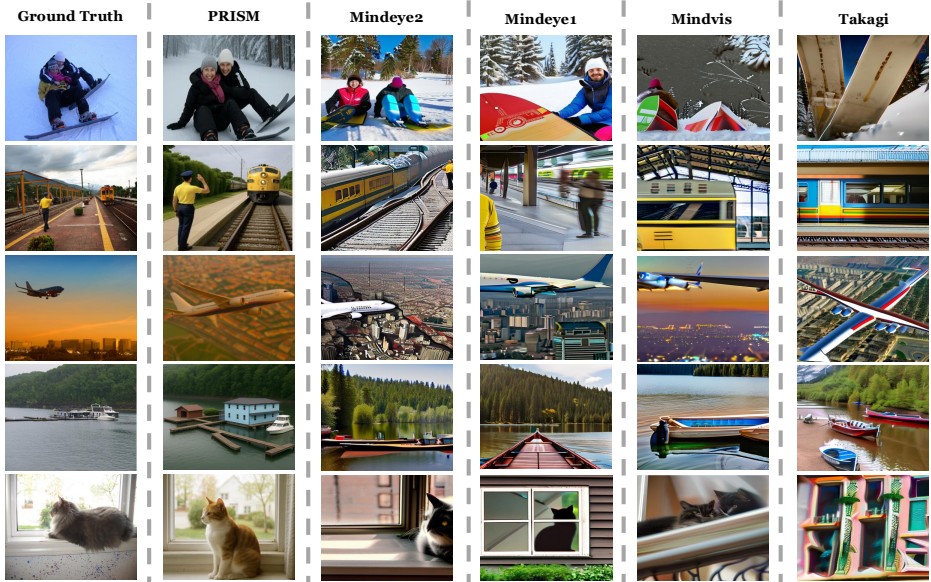

Figure 2: Reconstructed images from different methods. The first column shows the original viewed images. The rest of the columns show the reconstructed images from different methods.

of 60.54%, significantly outperforming state-of-the-art methods. This demonstrates that our reconstructions are not only visually faithful but also semantically meaningful.

### 4.3 ABLATION STUDY

In this subsection, we present ablation studies to justify our choice of text as the latent space and to evaluate the effectiveness of the object-centric diffusion and attribute/relationship search modules. Experiments are conducted on the NSD dataset, though the trend generalizes to other datasets.

We first compare reconstruction performance across three latent spaces: (1) language model embeddings (ours), (2) CLIP text embeddings (CLIP-Text), and (3) the image latent space of a diffusion model (LDM). As shown in Table 4, aligning fMRI signals to the language model text space consistently outperforms the other two spaces across all metrics. This demonstrates that textual representations alone can capture multiple levels of visual information, making text space a more brain-aligned and effective intermediate representation for fMRI-to-image reconstruction. Results for the remaining two datasets are provided in Table 7 of the Appendix.

Table 4: Reconstruction performance across three latent spaces. The best result in each column is highlighted in red. ↑ indicates higher is better and ↓ indicates lower is better.

| NSD | PixCorr ↑ | SSIM ↑ | LPIPS ↓ | CLIP ↑ | Inception V3 ↑ |
|---|---|---|---|---|---|
| **PRISM** | $0.3404_{\pm 0.05}$ | $0.4640_{\pm 0.02}$ | $0.5963_{\pm 0.02}$ | $0.9467_{\pm 0.03}$ | $0.9516_{\pm 0.03}$ |
| **Clip text** | $0.3208_{\pm 0.04}$ | $0.3725_{\pm 0.06}$ | $0.6611_{\pm 0.05}$ | $0.9197_{\pm 0.02}$ | $0.9011_{\pm 0.04}$ |
| **LDM** | $0.2090_{\pm 0.07}$ | $0.3727_{\pm 0.07}$ | $0.7502_{\pm 0.04}$ | $0.8602_{\pm 0.06}$ | $0.8925_{\pm 0.05}$ |

Next, we evaluate the effectiveness of the two proposed modules. Results are in Table 5. To evaluate the Object-centric Diffusion module, we compare against a variant (**w/o ObjC.**) that replaces object-level cross-attention with standard U-Net cross-attention. To assess the attribute/relationship search module, we test two variants that skip the search process and rely only on the initial keyword set: **w/o AttOpt.+Bst**, which fixes the prompt to the highest-scoring (best) keyword, and **w/o AttOpt.+Wst**, which fixes it to the lowest-scoring (worst) keyword.

Table 5: Effectiveness of the object-centric diffusion module and attribute/relationship search module on NSD data. The best result is highlighted in red.

| | PixCorr ↑ | SSIM ↑ | LPIPS ↓ |
|---|---|---|---|
| **PRISM** | $0.3404_{\pm 0.05}$ | $0.4640_{\pm 0.02}$ | $0.5963_{\pm 0.05}$ |
| **w/o ObjC.** | $0.3291_{\pm 0.06}$ | $0.4299_{\pm 0.06}$ | $0.6111_{\pm 0.05}$ |
| **w/o AttOpt.+Bst** | $0.3311_{\pm 0.04}$ | $0.4421_{\pm 0.01}$ | $0.6005_{\pm 0.02}$ |
| **w/o AttOpt.+Wst** | $0.3068_{\pm 0.05}$ | $0.4167_{\pm 0.02}$ | $0.6398_{\pm 0.05}$ |

Overall, removing or replacing the two modules consistently degrades performance across all metrics. Specifically, eliminating object cross-attention leads to notable declines that cannot be recov-

ered through prompt optimization, highlighting its essential role in reconstructing perceptually accurate images. Likewise, bypassing prompt optimization and using the best or worst initial attribute also reduces performance, indicating that the initial attributes alone are insufficient and underscoring the importance of prompt optimization in our model. The ablation study on the number of objects in our framework is shown in Section H of the Appendix.

## 4.4 KEYWORD SEARCH

**Case Study.** To better understand the keywords selected by our attribute/relationship search module, Table 6 presents the top-scoring keywords across different rounds of the $\varepsilon$-greedy search. The results show that, despite extensive exploration, the top-scoring keywords consistently converge toward spatially oriented relationships such as `Spatial Layout` and `Relative Position`. This suggests that: (1) descriptions emphasizing spatial information are most effective for guiding the diffusion model to accurately reconstruct images, as indicated by their highest LPIPS scores; and (2) these keywords also align well with fMRI data, as their CKA scores are no lower than those of the initial keywords, in accordance with the search constraints. This result is consistent with prior neuro-scientific findings showing that neural representations in the brain are sensitive to spatial arrangements and relative positions of objects (Zopf et al., 2018; Graumann et al., 2022). Therefore, we use `Spatial Layout` as the optimal keyword $a$ to generate structured descriptions for model training. Additional implementation details of the attribute/relationship search module can be found in Section I of the Appendix. To further demonstrate its robustness, we perform an experiment where all spatial-related keywords are deliberately excluded from the initial keyword set. The results, presented in Section J of the Appendix, show that the module is still able to identify the relevant spatial keywords, demonstrating the robustness and effectiveness of our search process.

**Neuroscientific Interpretation of the Optimal Keyword.** To explore why the optimal keyword `Spatial Layout` exhibits stronger alignment with fMRI signals, we conduct a gradient-based interpretability analysis to examine the neural correlations of different keyword types. Specifically, during inference on the test set, we identify predicted words associated with spatial relationships (e.g., directional terms) and functional attributes (e.g., actions or behaviors of objects), and compute the gradients of these tokens with respect to the input fMRI signals. Ideally, larger gradients indicate that specific voxel values within the fMRI contribute more strongly to predicting the given keyword, and are thus more relevant to that semantic category. By averaging these gradients across samples, we identify the ROIs that contribute most to each keyword category. For spatial relationships, we observe the strongest activation in the ROI 119 (HCP mmp1 atlas) (Glasser et al., 2016), which belongs to Presubiculum (PreS), a region linked to spatial memory (Dalton & Maguire, 2017; Boecker et al., 2024). In contrast, for functional attributes, the Ventromedial Visual Area 1 (VMV1, ROI 153 of HCP mmp1 atlas) shows the highest activation. Additionally, we compute the mean voxel intensity within each ROI and find that PreS exhibited a higher mean activation (0.0080) compared to VMV1 (0.0028). These findings suggest that the fMRI contains stronger activation in spatially relevant regions such as PreS, which may explain the higher alignment observed for spatial keywords. Nonetheless, we acknowledge that further validation from the neuroscience community is needed.

## 5 RELATED WORK

**FMRI-Image Reconstruction.** Early approaches leveraged linear models to decode fMRI signals into visual features (Kay et al., 2008; Takagi & Nishimoto, 2023). Later work explores deep generative models: approaches such as (Lin et al., 2022; Ozcelik et al., 2022; Goodfellow et al., 2020) map fMRI signals into the latent space of GANs for image reconstruction, while BraVL (Du et al., 2023) employs a multimodal VAE to jointly model relationships between brain activity and visual–linguistic features for neural decoding. With advances in vision–language models (Radford et al., 2021; Liang et al., 2024), more recent approaches map fMRI signals into CLIP's image-embedding space (Scotti et al., 2024; 2023) and then use diffusion models for image reconstruction (Rombach et al., 2022; Xu et al., 2023; Podell et al., 2023). Building on this diffusion-based direction, MindDiffuser (Lu et al., 2023) introduces a two-stage framework that first decodes semantic information and then aligns structural information with CLIP visual features decoded from fMRI. Unlike prior work that directly maps fMRI signals to joint text–image spaces (Wang et al.,

Table 6: Top-5 keywords scored by $1 - $ LPIPS before searching and after 10, 20, 30 search steps. The top-5 results remain unchanged after 30 search rounds. The search results indicate a clear preference for keywords related to spatial and positional relations, with most of the top-performing keywords in the final results containing the term '*spatial*'.

| Rank | Initial | Round 10 | Round 20 | Round 30+ |
|------|---------|----------|----------|-----------|
| #1 | Spatial Configuration | Spatial Arrangement | Spatial Organization | Spatial Layout |
| #2 | Positional Relation | Spatial Configuration | Spatial Structure | Spatial Patterns |
| #3 | Location Relation | Spatial Interaction | Spatial Arrangement | Spatial Organization |
| #4 | Descriptive Attribute | Positional Relation | Spatial Configuration | Relative Position |
| #5 | Inclusion Dependency | Feature Relation | Spatial Interaction | Spatial Relationships |

2024c; Quan et al., 2024), we compare multiple representation spaces and find that text embeddings from language models (Raffel et al., 2020) exhibit the strongest alignment with fMRI signals. This insight motivates our approach of reconstructing images via the embedding space of language models. Cross-subject decoding has also gained attention. MindTuner (Gong et al., 2025) introduces visual-fingerprint modeling using Skip-LoRA to improve cross-subject alignment and semantic correction. Psychometry (Quan et al., 2024) proposes an omni Mixture-of-Experts architecture that captures both inter-subject shared structure and subject-specific variability, combined with retrieval-enhanced inference for improved reconstruction. Wills Aligner (Bao et al., 2025) further advances multi-subject collaborative decoding by combining anatomical alignment with subject-guided Mixture-of-Brain-Expert adapters. Beyond static images, recent work has extended brain decoding to dynamic visual stimuli. NeuroClips (Gong et al., 2024) reconstructs such stimuli from fMRI by combining decoded keyframes with low-level perceptual flows to improve temporal consistency. BrainNEDS (Yeung et al., 2025) utilizes video diffusion models to disentangle static and dynamic components, offering insights into which visual features are prioritized by the brain.

**Diffusion models.** Diffusion models have become foundational in generative tasks like image creation and editing (Gal et al., 2022; Song et al., 2020), as well as text-to-image synthesis (Ruiz et al., 2023). To enhance control over generated content, ControlNet (Zhang et al., 2023) introduces high-level image features for controlling and GLIGEN (Li et al., 2023; Zhang et al., 2025) incorporates position-aware adapters for spatial grounded generation. Meanwhile, there are also training-free methods that adjust latent or attention maps during inference to guide outputs without additional training (Chen et al., 2024; Yang et al., 2024). In our work, we guide the diffusion process by modifying cross-attention layers during inference to integrate object-level descriptions derived from fMRI data for image reconstruction.

**Prompt Optimization.** Prompt optimization aims to discover effective textual prompts for LLMs without model fine-tuning. Gradient-based methods (Shin et al., 2020; Shi et al., 2022; Wen et al., 2023) update prompts using gradients or differentiable embeddings. Gradient-free approaches treat LLMs as black boxes, using heuristic search (Prasad et al., 2022; Pryzant et al., 2023), reinforcement learning (Deng et al., 2022; Zhang et al., 2022), or evolutionary strategies (Zhou et al., 2022; Yang et al., 2023; Guo et al., 2025). We designed our gradient-free prompt optimization based on beam search to optimize attribute keywords for black-box vision-language models.

# 6 CONCLUSION

In this work, we address the challenge of reconstructing visual stimuli from fMRI signals. Our analysis reveals that fMRI signals align more closely with the text space of language models than with vision-based or joint text–image representations, identifying text as a brain-aligned intermediate space. Building on this insight, we show that explicitly modeling the compositional structure of visual perception—capturing objects along with their attributes and relationships—further improves reconstruction quality. Guided by these findings, we develop **PRISM**, a framework that maps fMRI signals into a structured text space and incorporates two specialized modules: an object-centric diffusion module that generates images by composing individual objects, and an attribute/relationship search module that automatically discovers attributes and relationships aligned with neural activity. Experiments on real-world fMRI datasets demonstrate that PRISM reduces perceptual loss by up to 6% compared to prior methods, underscoring the power of structured text as a bridge between brain activity and image generation.

## 7 ETHICS STATEMENT

Our work does not involve human or animal subjects, personally identifiable data, or sensitive information. The datasets used are publicly available, and we follow their respective licenses. The methods and findings presented do not pose foreseeable risks of misuse, discrimination, or harm. We therefore believe our work raises no specific ethical concerns under the ICLR Code of Ethics.

## 8 REPRODUCIBILITY STATEMENT

Section 3 details the proposed framework and its design. Section 4 describes the datasets, baseline methods, and evaluation protocols used for comparison. Additional implementation details, including training procedures and hyperparameter settings, are provided in the Appendix. Upon acceptance of this paper, we will release our code on GitHub.

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

## A    A COMMON ERROR IN GENERATIVE MODELS

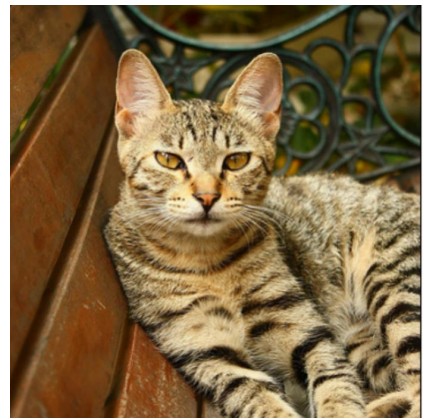 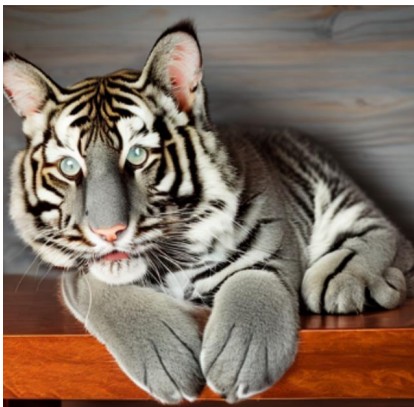

**Original Image**                    **Generated Image**

Figure 3: A Common Error in Generative Models. While the original image shows "a gray tiger-striped cat," the model incorrectly generates "a grey tiger," illustrating semantic distortion.

In this section, we present a common failure, attribute binding, encountered in generative models (especially for the diffusion model), where generative models misattribute visual properties to objects. Figure 3 compares original (left) and generated (right) images. The original depicts a gray tiger-striped cat on a wooden bench, while the generated version incorrectly shows a gray tiger instead of a cat. This issue arises because diffusion models usually rely on text encoders such as CLIP, which are known to lack the ability to capture complex linguistic structures (Yuksekgonul et al., 2022). Consequently, the diffusion process loses awareness of the bindings between objects and their attributes, leading to mismatched visual properties. This impairs fMRI-to-image reconstruction. To address this, we introduce a neuroscience-inspired, object-centric generation approach that improves reconstruction quality.

## B    PRELIMINARY: DIFFUSION MODEL

Diffusion models are a class of generative models that synthesize data by reversing a gradual noising process. Given a data point $\mathbf{H}_0$ (e.g., an image), the forward process perturbs it into Gaussian noise over $T$ time steps. The model then learns the reverse process to reconstruct samples from noise. The forward process is a Markov chain defined by:

$$q(\mathbf{H}_t \mid \mathbf{H}_{t-1}) = \mathcal{N}(\mathbf{H}_t; \sqrt{1 - \beta_t}\,\mathbf{H}_{t-1}, \beta_t \mathbf{I}), \quad t = 1, \ldots, T,$$

where $\{\beta_t\}_{t=1}^T$ is a predefined variance schedule. The model is trained to predict the noise $\boldsymbol{\epsilon}$ added to the input, using a neural network $\boldsymbol{\epsilon}_\theta$, by minimizing:

$$\mathcal{L} = \mathbb{E}_{\mathbf{H}_0, \boldsymbol{\epsilon}, t} \left[ \|\boldsymbol{\epsilon} - \boldsymbol{\epsilon}_\theta(\mathbf{H}_t, t)\|^2 \right].$$

Here, $\mathbf{H}_t = \sqrt{\bar{\alpha}_t}\,\mathbf{H}_0 + \sqrt{1 - \bar{\alpha}_t}\,\boldsymbol{\epsilon}$, with $\bar{\alpha}_t = \prod_{s=1}^t (1 - \beta_s)$, and $\boldsymbol{\epsilon} \sim \mathcal{N}(0, \mathbf{I})$. To guide the generation process with external information $\mathbf{C}$ (e.g., a text prompt), the denoising network is extended as:

$$\boldsymbol{\epsilon}_\theta(\mathbf{H}_t, t, \mathbf{C}).$$

Then, the training objective becomes:

$$\mathcal{L}_{\text{cond}} = \mathbb{E}_{\mathbf{H}_0, \boldsymbol{\epsilon}, t, \mathbf{C}} \left[ \|\boldsymbol{\epsilon} - \boldsymbol{\epsilon}_\theta(\mathbf{H}_t, t, \mathbf{C})\|^2 \right].$$

This formulation is widely used in text-to-image diffusion models, where $\mathbf{C}$ is the embedding of a textual description obtained from a pre-trained text encoder (e.g., CLIP). In practice, the condition $\mathbf{C}$ is incorporated into the U-Net via cross-attention modules.

In our work, we adopt the latent diffusion framework (Rombach et al., 2022), where the diffusion process is applied in the latent space of a pre-trained VAE, rather than directly in pixel space. Specifically, an input image $\mathbf{Y} \in \mathbb{R}^{H \times W \times 3}$ is first encoded by a VAE encoder into a compact latent representation $\mathbf{Z} \in \mathbb{R}^{h \times w \times c}$:

$$\mathbf{Z} = \text{Encoder}(\mathbf{Y}).$$

The diffusion process is applied on $\mathbf{Z}$, where the perturbed latent representation $\mathbf{Z}_T$ is obtained after T steps. The reversed denoising steps then generate a denoised latent $\hat{\mathbf{Z}}$ over T steps. The final image is reconstructed by the denoised latent:

$$\hat{\mathbf{Y}} = \text{Decoder}(\hat{\mathbf{Z}}).$$

This formulation greatly reduces computational cost while maintaining high-quality image generation and is particularly well-suited for conditioning on high-level semantic representations such as text or fMRI-derived embeddings.

## C   DATASET

In this subsection, we provide information about the three pre-processed datasets used for the fMRI-to-image reconstruction task: NSD (Allen et al., 2022), BOLD5000 (Chang et al., 2019), and GOD (Horikawa & Kamitani, 2017).

- **NSD** (Allen et al., 2022): The Natural Scenes Dataset (NSD) is a large-scale public fMRI dataset capturing brain responses of human participants viewing naturalistic stimuli from COCO images (Lin et al., 2014). The dataset includes scans for 30–40 hours across 30–40 separate sessions. During each session, participants viewed 750 images for 3 seconds each. Each image was presented three times across sessions, with most images unique to each subject, except for 1,000 shared images seen by all subjects. Following prior NSD reconstruction studies (Scotti et al., 2023; Takagi & Nishimoto, 2023), we adopt the standardized train/test split, where the shared images serve the test set. Consequently, the training set for each subject contains 8,859 image stimuli and 24,980 fMRI trials, while the test set includes 982 image stimuli and 2,770 fMRI trials.

- **BOLD5000** (Chang et al., 2019): The BOLD5000 dataset is a publicly available fMRI dataset capturing brain activity as subjects viewed a series of images. It contains 4,916 unique images, including 2,000 from the COCO dataset and 1,916 from ImageNet (Deng et al., 2009). Each image was presented as a visual stimulus in individual trials. Of these, 4,803 images were shown once, while 113 images were repeated three or four times across trials, resulting in a total of 5,254 stimulus trials. We follow the standardized train/test split used in prior BOLD5000 reconstruction studies (Chen et al., 2023; Wang et al., 2024c). Specifically, the training set includes trials with non-repeated image stimuli, comprising 4,803 samples, while the test set consolidate repeated image stimulus trials into 113 samples.

- **GOD** (Horikawa & Kamitani, 2017): The Generic Object Decoding (GOD) is a public dataset developed for fMRI based decoding. It aggregates fMRI data gathered through the presentation of images from 200 representative object categories, originating from ImageNet. We follow the standardized train/test set split employed in existing GOD image reconstruction studies (Sun et al., 2023) and get 1200 training samples and 50 test samples.The Generic Object Decoding (GOD) dataset is a publicly available fMRI dataset designed for decoding object representations. It includes fMRI data collected during the presentation of images from 200 representative object categories sourced from ImageNet. Following the standardized train/test split used in prior GOD reconstruction studies (Sun et al., 2023), we use 1,200 training samples and 50 test samples.

## D  BASELINES

In this section, we provide details about the baselines used in our experiments.

- Takagi & Nishimoto (Takagi & Nishimoto, 2023): This baseline maps fMRI signals to the latent space of a pre-trained VAE within a diffusion model using linear regression, enabling image reconstruction. The method combines image latent representations with text embeddings extracted from a CLIP text encoder, both mapped from fMRI signals in higher (ventral) visual cortex regions, to improve reconstruction quality. For a fair comparison, we adapt this approach to the Diffusion 2.1 pipeline by retraining the linear regression to map fMRI signals to both the VAE latent space and the textual conditioning used in Diffusion 2.1.

- Mindvis (Chen et al., 2023): This baseline uses a self-supervised representation of fMRI data using masked modeling within a high-dimensional latent space in an encoder–decoder framework. The learned representation is then projected into the conditioning space of LDM by fine-tuning the model. For fair comparison, we adapt the image generator of this approach to Diffusion 2.1 by fine-tuning it with the learned projection module, following the strategy outlined by the original authors.

- Mindeye (Scotti et al., 2023): This model proposed two modules to map the fMRI signal to the CLIP image space. Specifically, the model first uses contrastive learning to align fMRI signals with image embeddings. Second, the paper trains a diffusion prior to reconstructing images from these embeddings via mapping brain activity into CLIP image space, enabling the generation of images that closely resemble the original stimuli. To adapt the method for fair comparison, we replace the Versatile Diffusion with Diffusion 2.1.

- MindBridge (Wang et al., 2024b): This approach unifies heterogeneous voxel dimensions using adaptive max pooling and employs a cyclic fMRI reconstruction mechanism to align neural responses across subjects in a common semantic space. To adapt the method for fair comparison, we replace the Versatile Diffusion with Diffusion 2.1.

- NeuralDiffuser (Li et al., 2025): This paper proposes a visual feature–guided reconstruction method that decodes multiple layers of CLIP's visual encoder from fMRI and uses these features as gradient-based cues during the early stages of reverse diffusion, enabling bottom-up detail refinement. It further introduces momentum alignment to reduce the distribution shift between training and testing embeddings. For fair comparison, we use Diffusion 2.1 as the generative backbone.

- Mindeye2 (Scotti et al., 2024): This method trains multiple MLPs to project fMRI signals from all subjects into a shared representation space, followed by training a diffusion prior to map these representations into the CLIP image embedding space. The final image is then reconstructed using a pre-trained SDXL (Podell et al., 2023). To adapt this method to our setting, we replace the generative backbone with Diffusion 2.1.

## E  TRAINING DETAILS

In this section, we provide the training details of our model. Our model is implemented with Pytorch and trained on two NVIDIA-L40 GPUs with 48GB of memory. We use T5 as the language model to generate the object-level descriptions. For NSD data, we train the model for 80 epochs: 60 epochs for MLP training ($E_{\text{MLP}}$) with a learning rate $\text{lr}_1 = 1 \times 10^{-5}$, followed by 20 epochs of joint training, where we continue training the MLP and fine-tune the T5 model ($E_{\text{T5}}$) using a learning rate $\text{lr}_2 = 5 \times 10^{-7}$. For BOLD5000, we set $E_{\text{mlp}} = 50$, $\text{lr}_1 = 1e - 5$, $E_{\text{T5}} = 5$, and $\text{lr}_2 = 1e^{10-8}$. For GOD, we set $E_{\text{mlp}} = 40$, $\text{lr}_1 = 1e - 5$, $E_{T5} = 5$, and $\text{lr}_2 = 5e^{-9}$. For image reconstruction at inference time, we set the blending ratio $\beta = 0.5$ and the denoising step as 40 for all the datasets. We use `GPT 4o-mini` to generate the object-centric descriptions with the prompt shown in Section K. For images that do not have a caption, we first use `GPT-4o-mini` to generate a short caption and then use our prompt to generate the object-centric description.

Table 7: Reconstruction performance across three latent spaces. The best result in each column is highlighted in red. ↑ indicates higher is better and ↓ indicates lower is better.

| | PixCorr ↑ | SSIM ↑ | LPIPS ↓ | CLIP ↑ | Inception V3 ↑ |
|---|---|---|---|---|---|
| **BOLD5000** | | | | | |
| **Ours** | $0.2315_{\pm0.01}$ | $0.5341_{\pm0.02}$ | $0.6198_{\pm0.02}$ | $0.7720_{\pm0.03}$ | $0.6601_{\pm0.07}$ |
| **Clip text** | $0.2000_{\pm0.06}$ | $0.4885_{\pm0.06}$ | $0.6520_{\pm0.04}$ | $0.7265_{\pm0.04}$ | $0.6199_{\pm0.04}$ |
| **LDM** | $0.1622_{\pm0.03}$ | $0.4300_{\pm0.02}$ | $0.7894_{\pm0.08}$ | $0.7025_{\pm0.08}$ | $0.5590_{\pm0.05}$ |
| **GOD** | | | | | |
| **Ours** | $0.2571_{\pm0.01}$ | $0.5200_{\pm0.02}$ | $0.6213_{\pm0.01}$ | $0.8567_{\pm0.05}$ | $0.8428_{\pm0.06}$ |
| **Clip text** | $0.2200_{\pm0.05}$ | $0.4682_{\pm0.04}$ | $0.6827_{\pm0.04}$ | $0.8120_{\pm0.05}$ | $0.7602_{\pm0.04}$ |
| **LDM** | $0.1900_{\pm0.02}$ | $0.3999_{\pm0.07}$ | $0.7100_{\pm0.02}$ | $0.7099_{\pm0.01}$ | $0.7484_{\pm0.03}$ |

# F  CASE STUDY ON OBJECT-LEVEL DESCRIPTIONS FOR IMAGE RECONSTRUCTION

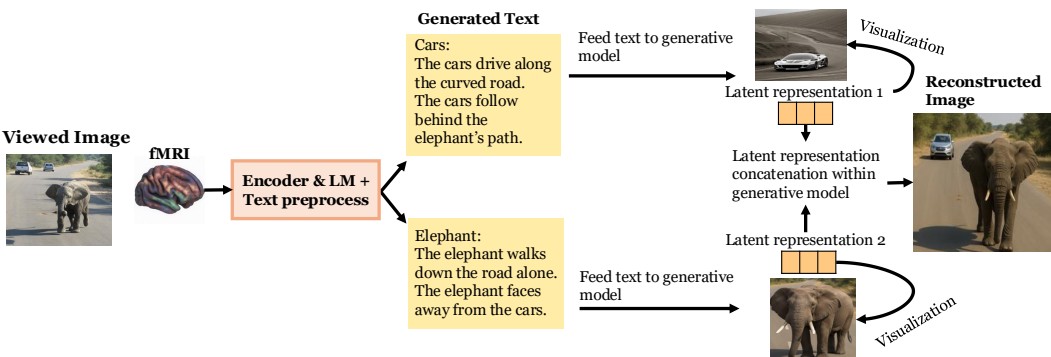

Figure 4: Reconstructed images using object-level descriptions.

In this section, we present a case study illustrating how object-level descriptions are used during the reconstruction process, as shown in Figure 4. In the given example, the generated text is segmented based on object instances: one segment describes the cars ("The cars drive along the curved road...") and the other describes the elephant ("The elephant walks down the road alone..."). Each object description is independently fed into a diffusion-based generative model to produce object-centric latent representations. These representations are then spatially aligned and concatenated within the generative model to produce the final reconstructed image. This modular process helps mitigate common generation errors, such as attribute binding.

# G  RECONSTRUCTION PERFORMANCE ACROSS DIFFERENT LATENT SPACES

We report the ablation studies to justify our choice of text as the latent space on BOLD5000 and GOD. As shown in Table 7, aligning fMRI signals to the language model text space (our method) consistently outperforms alignment to the other two spaces across all metrics. This supports our core contribution: textual representations alone are sufficient to capture both high-level semantic and low-level visual information, and text space provides a more brain-aligned and effective intermediate representation for fMRI-to-image reconstruction.

Table 8: Comparison of the number of objects in our framework. Results are reported using PixCorr, SSIM, LPIPS, CLIP, and Inception V3 metrics. The best result in each column is highlighted in red. ↑ indicates higher is better and ↓ indicates lower is better.

| | PixCorr ↑ | SSIM ↑ | LPIPS ↓ | CLIP ↑ | Inception V3 ↑ |
|---|---|---|---|---|---|
| Ours (Two Objs) | 0.3404 $_{\pm 0.05}$ | 0.464 $_{\pm 0.02}$ | 0.5943 $_{\pm 0.02}$ | 0.9467 $_{\pm 0.03}$ | 0.9516 $_{\pm 0.03}$ |
| One Obj | 0.3355 $_{\pm 0.05}$ | 0.4532 $_{\pm 0.04}$ | 0.6014 $_{\pm 0.04}$ | 0.9344 $_{\pm 0.02}$ | 0.9342 $_{\pm 0.01}$ |
| Four Objs | 0.3202 $_{\pm 0.03}$ | 0.4469 $_{\pm 0.04}$ | 0.6284 $_{\pm 0.02}$ | 0.9400 $_{\pm 0.05}$ | 0.9322 $_{\pm 0.05}$ |

## H ANALYSIS ON THE NUMBER OF OBJECTS IN OUR FRAMEWORK

In our framework, we fix the number of objects per image to two and assign a separate MLP to each. We learn to assign each object a location label from a predefined set of spatial positions (e.g., left/right or top/bottom). This fixed assignment of each object to a dedicated MLP, along with the predefined spatial labeling scheme, is used consistently during both training and inference. At inference time, each MLP independently encodes fMRI signals for one object, and the language model generates a structured description for each. These descriptions are then passed to the object-centric diffusion model, which generates object images independently and places them into their corresponding spatial positions to form the final image.

We conduct an experiment to determine the optimal number of objects (MLPs) in our framework, and the results on the NSD dataset are reported in Table 8. The results show that setting the number of objects per image to two yields the best performance. We further present case studies for setting $m = 2$ and $m = 4$ in Figures 5 and 6, respectively. We observe that for more complex images (i.e., those containing four objects), setting $m = 2$ encourages the VLM to describe the first object as a group of similar instances, rather than generating isolated descriptions for each object. As shown in Figure 5, the individual players are grouped and described as "three players." In contrast, setting $m = 4$ prompts the VLM to generate object-level descriptions (e.g., generating one description per player, as shown in Figure 6), leading the generative model to produce distinct latent representations for each object. These representations are then concatenated by spatial location and passed to the pre-trained diffusion model. However, when too many object-level features are introduced, the diffusion model tends to omit objects during the denoising process Liu et al. (2024); Wang et al. (2023), resulting in incomplete reconstructions—for example, with one of three players missing in the final output (Figure 5).

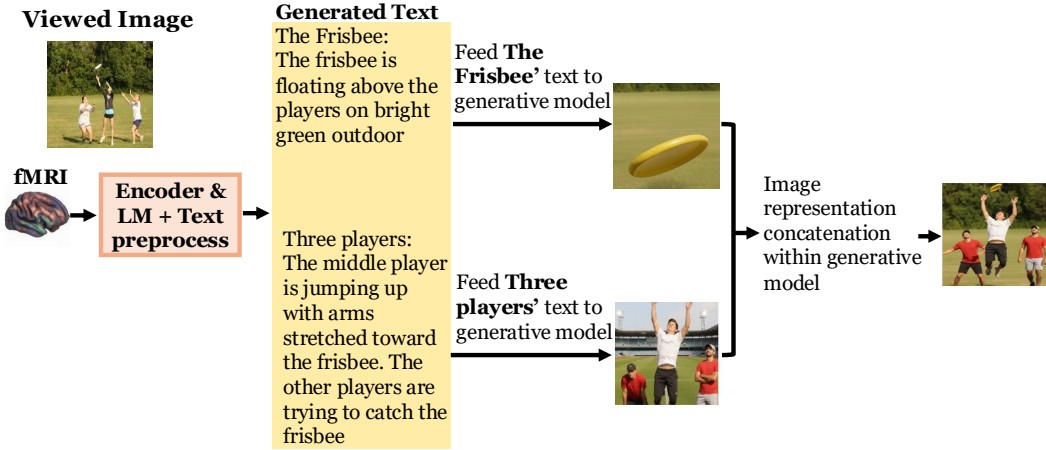

Figure 5: Reconstructed images using two objects.

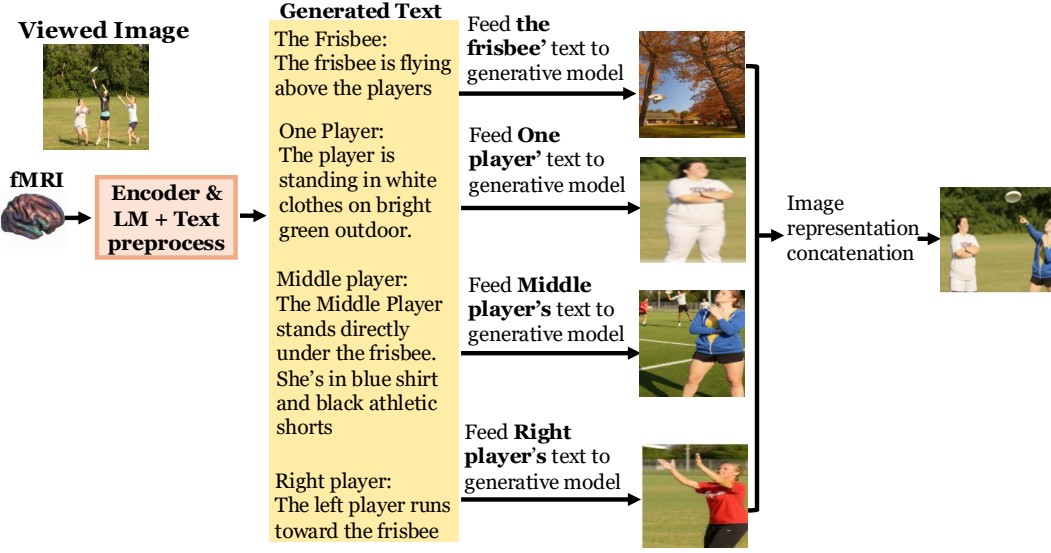

Figure 6: Reconstructed images using four objects.

## I ATTRIBUTE/RELATIONSHIP OPTIMIZATION DETAILS

The detailed algorithm used to solve the prompt optimization problem in Equation (3) is provided in Algorithm 1. In our experiments, we adopt the scoring function $\mathcal{S}(\mathbf{Y}_1, \mathbf{Y}_2) = 1 - \mathrm{LPIPS}(\mathbf{Y}_1, \mathbf{Y}_2)$. The CKA threshold $\beta$ is initialized to the minimum CKA score among the initial candidates. We set $\varepsilon = 0.5, k_1 = 8, k_2 = 2$ and search for $T = 40$ rounds. We randomly sampled 667 images from the training set of **NSD** (Allen et al., 2022) for prompt optimization. While this subset is used for efficiency, our method is applicable to the full training set and generalizes to other settings. We use GPT-4o-mini as the VLM and the LLM that generates new keywords. We use CLIP-text (Radford et al., 2021) as $\mathrm{LM_{ENC}}$, and Stable Diffusion 2.1 (Pernias et al., 2023; Rombach et al., 2022) as the diffusion model.

---

**Algorithm 1** $\varepsilon$-Greedy Prompt Optimization

**Input:** Training set $\mathcal{X}^{\mathrm{train}}, \mathcal{Y}^{\mathrm{train}}$; Initial keyword set $\mathcal{A}$; Search rounds $T$; Parameters $\varepsilon, k_1, k_2$
**Initialize:** Threshold $\beta \leftarrow \min_{a \in \mathcal{A}} \mathrm{CKA}(\mathbf{X}, \mathbf{K}^a)$
**for** $t = 1$ to $T$ **do**
    **Filter:** $\mathcal{A} \leftarrow \{a \in \mathcal{A} \mid \mathrm{CKA}(\mathbf{X}, \mathbf{K}^a) > \beta\}$
    **Sort:** Rank $a \in \mathcal{A}$ in descending order by $\sum_{i=1}^{N} \mathcal{S}(\mathbf{Y}_i, \mathrm{Diff}(\mathrm{VLM}(\mathbf{Y}_i, \mathcal{P}(a))))$
    **if** random() $< \varepsilon$ **then**
        **Sample:** $\mathcal{S} \leftarrow \mathrm{RandomSample}(\mathcal{A}, k_1)$       % Randomly sample $k_1$ keywords from $\mathcal{A}$
    **else**
        **Select:** $\mathcal{S} \leftarrow \mathrm{Top}(\mathcal{A}, k_1)$              % Select top-$k_1$ keywords from $\mathcal{A}$
    **end if**
    **Generate:** Use LLM to synthesize $k_2$ new keywords $\mathcal{A}_{\mathrm{new}}$ based on $\mathcal{S}$
    **Update:** $\mathcal{A} \leftarrow \mathcal{A} \cup \mathcal{A}_{\mathrm{new}}$
**end for**
**Output:** $\arg\max_{a \in \mathcal{A}} \sum_{i=1}^{N} \mathcal{S}(\mathbf{Y}_i, \mathrm{Diff}(\mathrm{VLM}(\mathbf{Y}_i, \mathcal{P}(a))))$

---

The search is initialized with six widely-used keywords describing object attributes and relationships: Semantic Relationship (Johnson et al., 2015), Positional Relationship (Lu et al., 2016; Haldekar et al., 2017), Functional Attributes (Zhu et al., 2015), Action Attributes (Lu et al., 2016), Visual Attributes (Farhadi et al., 2009), and Part–Whole Relationship (Lu et al., 2016). For each type, we use GPT-4o to generate four synonymous keywords, resulting in an initial pool of 24 candidate keywords. Figure 7 reports the LPIPS scores of all initial and subsequently discovered keywords.

## J  KEYWORD SEARCH WITH DIFFERENT INITIAL SETS

To further demonstrate the effectiveness of our attribute/relationship search module, we conduct an additional experiment where we excluded spatial-related keywords from the initial keyword set. We then evaluated whether our search procedure could still find spatially relevant terms. As shown in Table 9, the module successfully identifies spatial keywords under this constraint, demonstrating the robustness and effectiveness of our search process.

Table 9: Top-5 discovered keywords across search rounds.

| Rank | Round 0 | Round 10 | Round 20 | Round 30 |
|------|---------|----------|----------|----------|
| #1 | Textural Attribute | Spatial Relationship | Spatial Arrangement | Spatial Arrangement |
| #2 | Conceptual Linkage | Textural Attribute | Proximity Dynamics | Proximity Relation |
| #3 | Action Interaction | Visual Harmony | Spatial Relationship | Proximity Dynamics |
| #4 | Chromatic Feature | Emotional Resonance | Color Arrangement | Spatial Relationship |
| #5 | Attribute Description | Conceptual Linkage | Hue Contrast | Color Arrangement |

## K  PROMPTS USED BY **PRISM**

---

**Prompt in generating structured descriptions**

Given the image and caption, first describe the background color style of the image with 3-5 words. Second, detect the TWO most important objects in the image. Then, describe each of the objects and their relationship using: **{keyword}** with TWO sentences. For each sentence, use 5-10 words and as easy as possible.

Then, detect the absolute position of the two objects in the image, and select from [right, left, top, bottom]. "left" and "right" should appear together for horizontal objects, and "top" and "bottom" should appear together for vertical objects. DO NOT mix.

Example:

> ### Background color style: Grayscale urban.
>
> ### The Man [left]
> 1. The man is standing near the sidewalk edge. The Man is close to the building wall.
>
> ### The Suitcase [right]
> 1. The suitcase is beside the man's foot. The Suitcase is placed on the street's curved edge.

Now, given the image I uploaded and the caption "**{caption}**", detect the two most important objects with absolute position, describe them using **{keyword}** with EXACTLY the example format:

---

**Prompt in attribute/relationship optimization**

System prompt: You are a helpful brainstormer. Given a list of keywords, generate **{gen_num}** related or similar keywords. Respond with a comma-separated list of keywords.

User prompt: keyword: **{keywords}**

---

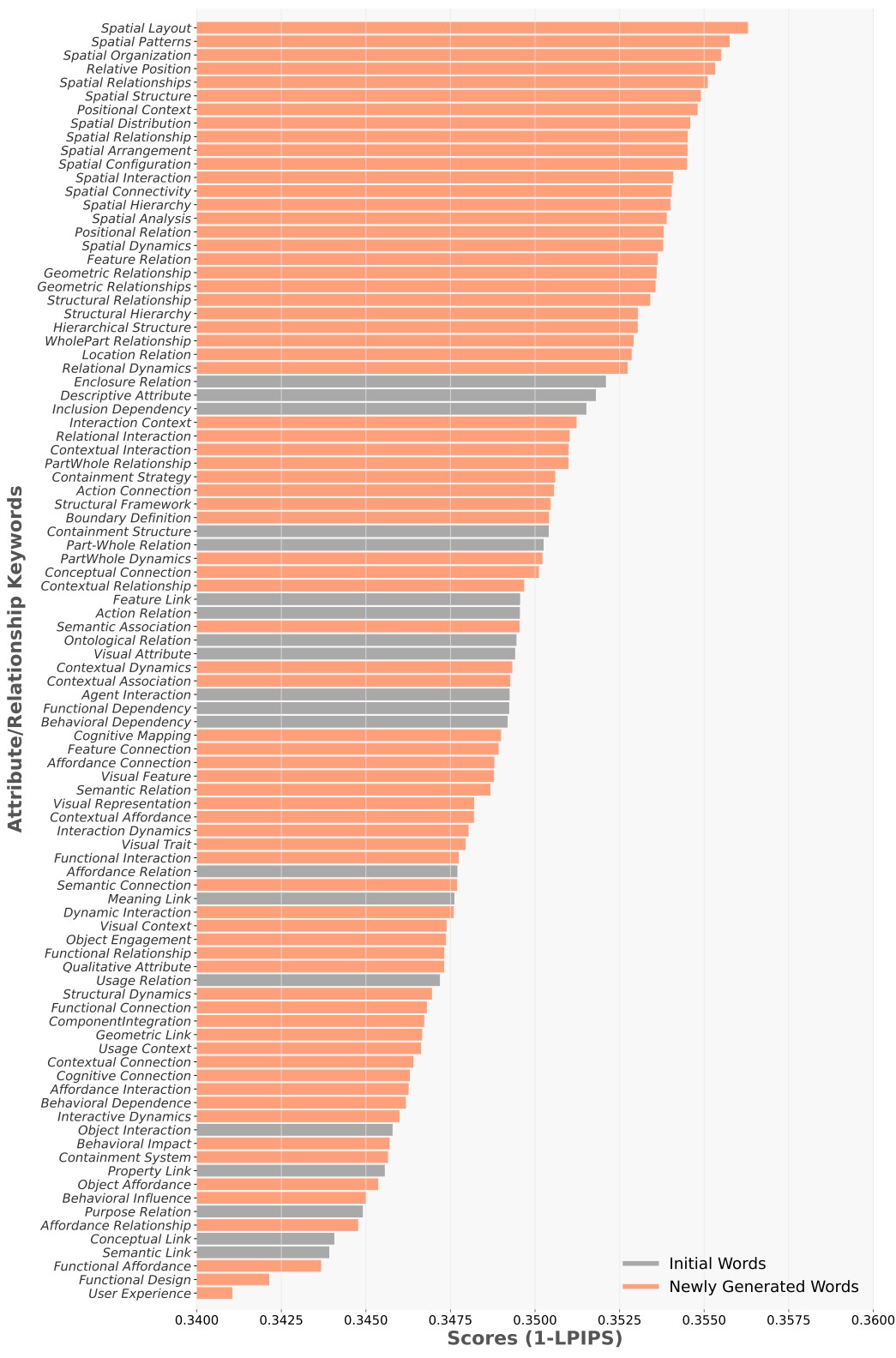

Figure 7: Scores of all keywords during the prompt search. Among the top-10 keywords, the most frequent keyword is 'spatial'.

## L   THE USE OF LARGE LANGUAGE MODELS (LLMs)

We declare that Large Language Models (LLMs) were confined to peripheral tasks and had no influence on the methodology, results interpretation, or theoretical insights of this work. Specifically, they were used for (i) generating training datasets required for our experiments and (ii) grammar correction and minor word-level refinements. All language edits were carefully reviewed by the authors to ensure that no hallucinations were introduced and that the text faithfully reflects the original intent. The technical development, experimental design, analysis, and conclusions presented here are entirely the work of the authors.

