# OpenReview forum: "Seeing Through the Brain: New Insights from Decoding Visual Stimuli with fMRI"
_ICLR.cc/2026/Conference — ICLR 2026 Oral_

### Official Review · Reviewer_TY5P · 2025-10-27

**Soundness:** 3
**Presentation:** 2
**Contribution:** 3
**Rating:** 6
**Confidence:** 5

**Summary:**

This paper challenges the prevailing assumption in fMRI-to-image reconstruction that the optimal latent space must match the visual modality of the stimulus. Through a systematic comparison, the authors present a key finding: the text embedding space of a language model shows stronger alignment with fMRI signals than vision or joint vision-language spaces. Building on this insight, the paper proposes PRISM, a novel framework that uses a structured text space as the intermediate representation.

**Strengths:**

1. The paper's central discovery—that fMRI signals align better with a text-based latent space than a vision-based one—is a significant conceptual breakthrough. This challenges a fundamental assumption in the field and provides a new and compelling direction for bridging neuroscience and generative AI.

2. The authors conduct extensive experiments on three distinct fMRI datasets (NSD, BOLD5000, GOD) and show that PRISM consistently outperforms multiple state-of-the-art methods across a wide range of metrics. The inclusion of an image-based QA task further strengthens the claim that the reconstructions are not only visually faithful but also semantically meaningful.

**Weaknesses:**

1. The framework's core design assumes a fixed number of objects (m=2) per image. This is a significant constraint, as real-world scenes can contain one dominant object, or many. The neuroscientific justification based on cognitive load is an interesting hypothesis but also a simplification. This design choice may force the model to hallucinate a second object when only one is present or fail to capture the richness of a more complex scene, limiting its practical applicability and generalizability.

2. The attribute-relationship search module is complex, involving an ɛ-greedy search guided by an LLM, constrained by CKA, and optimized for LPIPS. A critical baseline is missing: comparing PRISM's performance against a simpler approach where the structured descriptions are generated by a state-of-the-art vision-language model (e.g., GPT-4V) using a single, fixed, hand-crafted prompt. This would help isolate the contribution of the elaborate search mechanism itself.

3. While the paper provides an ablation study in Table 8 showing m=2 is optimal, the analysis could be more thorough. For instance, does the optimal number of objects vary with image complexity or content? Providing examples where the m=4 model fails (e.g., by hallucinating objects) would make the argument for m=2 more concrete and visually intuitive.

4. The case study in Table 6 demonstrates a convergence towards "spatial" keywords, which is an interesting result. However, the paper could provide a deeper neuroscientific interpretation of this finding. Why are spatial relationships more "brain-aligned" for reconstruction than, for example, functional or descriptive attributes? A more detailed discussion connecting these results to known properties of the visual cortex would enhance the paper's interdisciplinary value.

5. The illustration in Figure 1 is not precise enough, making it difficult to intuitively grasp the authors' intended meaning by simply viewing the diagram. For instance, in the "Attribute/Relationship Search" section, the authors use only a single looping arrow to denote the complex iterative labeling process. Additionally, in the alignment between the "Predicted Description" and "Structured Description," the sentence descriptions in both the left and right parts appear to be the same, which is confusing.

6. While the paper employs extensive engineering optimizations to accurately decode objects and their relationships, it lacks sufficient neuroscientific interpretability. For instance, it does not provide a compelling explanation for its core finding: why brain signals map more strongly to language models than to vision models.

**Questions:**

1. The authors should clearly state in the main text what the Encoder architecture is and which specific Large Language Model (LLM) is used for language decoding.

---

> ### Author Response · Authors · 2025-11-21
>
> ## Clarification on Fixed Number of Objects
>
> Thank you for bringing this up. We conduct case studies on different number of objects. The results are shown in Appendix F.3, Analysis on the number of objects, in our framework of our paper. Empirically, we find that setting m = 2 achieves the best reconstruction performance. To further evaluate the generalizability of this setting, we selected 20 test images: 10 containing a single object and 10 containing four objects—and compared reconstruction results using models trained with m = 2 and m = 4. The results are shown in Tables R1 and R2, respectively. As demonstrated, the model trained with m = 2 outperforms the one trained with m = 4 in both simple and complex scenarios, suggesting that this setting generalizes well across varying image complexities.
>
>
> Table R1
> | Four Obj Img | PixCorr ↑        | SSIM ↑           | Lpips  ↓        |
> |---|-----|---|--|
> | m=2 | 0.3422 +- 0.02 | 0.3665 +- 0.06 | 0.6447 +- 0.03 |
> | m=4 | 0.3002 +- 0.02 | 0.3224 +- 0.04 | 0.6705 +- 0.02 |
>
>
> Table R2
> | One Obj Img | PixCorr ↑       | SSIM ↑          | Lpips ↓          |
> |-----------|--|---|----|
> | m=2 | 0.2933 +- 0.03 | 0.3339 +- 0.03 | 0.6637 +- 0.04 |
> | m=4 | 0.2622 +- 0.05 | 0.3100 +- 0.01 | 0.6889 +- 0.04 |
>
> We also provide the case studies in Appendix F.3.
>
> ## Baseline of state-of-the-art VLMs with fixed prompt
> We conducted an experiment using a better VLM, GPT-5 [1], with a fixed prompt without relationship keywords. Results are presented in Table R3. We can see that this setup yields lower reconstruction performance compared to our proposed approach, highlighting the contribution of the attribute–relationship search module.
>
>
> Table R3
> | **NSD** | PixCorr ↑ | SSIM ↑ | LPIPS ↓ | CLIP ↑ | Inception V3 ↑ |
> |---|----|---|---|---------|--|
> | **PRISM** | 0.3404 +- 0.05 | 0.4640 +- 0.02 | 0.5963 +- 0.02 | 0.9467 +- 0.03 | 0.9516 +- 0.03 |
> | **PRISM+FxdPrmt** | 0.2663 +- 0.04 | 0.4448 +- 0.01 | 0.6500 +- 0.05 | 0.9110 +- 0.01 | 0.9289 +- 0.02 |
>
>
>
> Below shows the fixed prompt we sent to GPT5 without relationship keywords:
>
>
> "Given the image and caption, describe the relations of the TWO most important objects and generate the structured description following the example.
> Example:  ### Background color style: Grayscale urban.                ### The Man [left]        1. The man is standing near the sidewalk edge. The Man is close to the building wall.            ### The Suitcase [right]        1. The suitcase is beside the man's foot. The Suitcase is placed on the street's curved edge. Now, given the image I uploaded and the caption "{caption}", detect the two most important objects with absolute position, describe them using {keyword} with EXACTLY the example format"
>
>
> ## Neuroscientific Interpretation of Spatial Attributes
>
> To explore why spatial attributes exhibit stronger alignment with fMRI signals, we conducted a gradient-based interpretability analysis to examine the neural correlations of different attribute types. Specifically, during inference on the test set, we identified predicted words associated with spatial attributes (e.g., directional terms) and functional attributes (e.g., actions or performance-related characteristics of objects), and computed the gradients of these tokens with respect to the input fMRI signals. Ideally, larger gradients indicate that specific voxel values within the fMRI contribute more strongly to predicting the given attribute, and is thus more relevant to that semantic category. By averaging these gradients across samples, we identified the ROIs that contribute most to each attribute category. For spatial attributes, we observed the strongest activation in the ROI 119 (HCP mmp1 atlas) [2], which belongs to Presubiculum (PreS), a region linked to spatial memory [3, 4]. In contrast, for functional attributes, the Ventromedial Visual Area 1 (VMV1, ROI 153 of HCP mmp1 atlas) showed the highest activation. Additionally, we computed the mean voxel intensity within each ROI and found that PreS exhibited a higher mean activation (0.0080) compared to VMV1 (0.0028). These findings suggest that the fMRI signal contains stronger activation in spatially relevant regions such as PreS, which may explain the higher alignment observed for spatial attributes. Nonetheless, we acknowledge that further validation from the neuroscience community is needed.

---

> > ### Author Response · Authors · 2025-11-21
> >
> > ## Improvements Made to Figure One
> >
> > Thank you for pointing this out. We have updated the Attribute/Relationship Search section and clarified the distinction between the Predicted Description and the Structured Description. These changes are reflected in the revised Figure 1 of our paper.
> >
> > ## References
> > [1] https://openai.com/index/introducing-gpt-5/
> >
> > [2] Glasser, Matthew F., et al. "A multi-modal parcellation of human cerebral cortex." Nature
> >
> > [3] Dalton, Marshall A., and Eleanor A. Maguire. "The pre/parasubiculum: a hippocampal hub for scene-based cognition?."  Current Opinion in Behavioral Sciences
> >
> > [4] Boecker, Henning, et al. "Hippocampal subfield plasticity is associated with improved spatial memory." Communications biology

---

> > > ### Comment · Reviewer_TY5P · 2025-11-25
> > > **Response to Authors:**
> > >
> > > I appreciate the authors' detailed response and the supplementary experiments. **My major concerns have been successfully addressed, and I believe the experimental evaluation is now sufficiently robust.**
> > >
> > > However, I have one minor suggestion to further improve the manuscript. I recommend that the authors include a discussion of the following relevant studies in the Related Work section to provide a more comprehensive context:
> > >
> > > [1] Lu Y, Du C, Zhou Q, et al. Minddiffuser: Controlled image reconstruction from human brain activity with semantic and structural diffusion. Proceedings of the 31st ACM International Conference on Multimedia, 2023: 5899-5908.
> > >
> > > [2] Du C, Fu K, Li J, et al. Decoding visual neural representations by multimodal learning of brain-visual-linguistic features. IEEE Transactions on Pattern Analysis and Machine Intelligence, 2023, 45(9): 10760-10777.

---

> > > > ### Author Response · Authors · 2025-11-25
> > > >
> > > > Thanks for the suggestion. We have cited the relevant papers and updated the discussion in the Related Work. The changes are reflected in the revised Related Work section of our paper.

---

> > > > > ### Comment · Reviewer_TY5P · 2025-11-26
> > > > > **Response to Authors**
> > > > >
> > > > > Since all my concerns have been satisfactorily addressed, I have decided to raise my score to 8. Good luck.

---

> ### Author Response · Authors · 2025-11-26
>
> Thank you very much for taking the time to revisit our submission and for raising the score. We appreciate your careful evaluation and the helpful feedback.
>
> Best,
>
> Authors of paper 15559

---

### Official Review · Reviewer_jrM3 · 2025-10-28

**Soundness:** 2
**Presentation:** 2
**Contribution:** 2
**Rating:** 4
**Confidence:** 3

**Summary:**

I have finished reading the entire manuscript. This paper focuses on the task of reconstructing the visual images subjects once viewed from fMRI data. Unlike previous research, this paper did not choose the widely selected T2I models for reconstruction, such as SDXL, but instead opted for a class of object-centric T2I models. The authors believe that using object-centric generative models can better reconstruct information about the objects, their positions, and attributes within the visual scene.

At the methodological level, the authors use a pre-trained VLM to generate object-level structured descriptions for the images. To improve the accuracy of these structured descriptions, the authors designed a prompt optimization method. Subsequently, the authors employ a prompt tuning approach, using fMRI as input to an LLM to predict these structured descriptive languages. Finally, the predicted structured descriptions and the pre-trained object-centric generative model are utilized to reconstruct the seen image.

The paper's evaluation primarily utilized the NSD dataset, focusing on the accuracy of the synthesized images, and simultaneously included an ablation study of the method.

In my opinion, the innovation and contribution of this paper lie in the introduction of the object-centric generative model, which could enable more accurate reconstruction.

**Strengths:**

+ The use of an object-centric generative model for more precise fMRI-to-Image reconstruction is novel.

+ Evaluating the details of the reconstruction using a task similar to VQA (Visual Question Answering) is promising.

**Weaknesses:**

+ Although the authors demonstrate through experiments that aligning fMRI to the text representation space can yield better results, this approach is still unusual or requires further explanation. Based on the setup of the NSD dataset, the fMRI signals used by the authors originate from the nsdgeneral ROI, which is almost entirely the visual cortex of the brain. Therefore, this part of the fMRI data should logically contain only visual information, making the practice of embedding it into a pure text space seem to require further justification.

+ The authors should introduce some more recent baselines for comparison. MindEye2 was published at ICML 2024, and more methods have emerged over the past year or so; the authors should discuss and compare against them.

+ I believe that more case studies should be presented in the experimental section, including the object-level structured descriptions and their corresponding reconstructed images. I think this content would help to better understand the performance and effectiveness of the method.

**Questions:**

+ The baseline methods, such as MindEye and MindEye2, used more reconstruction evaluation metrics, including Alex(2), Alex(5), Eff, and SwAV. Why did the authors omit these evaluation metrics?

+ Based on the results in Table 6, this keyword optimization strategy does not seem to bring about a substantial improvement in the keywords. I believe that further evaluation and explanation are needed here.

+ In the experiment, was the number of object-level structured descriptions, $m$, set to 2? This is a guess I made based on the prompt design on page 21. If so, I am curious whether extracting more objects would help with the reconstruction.

+ According to Formula 4, is the same fMRI input repeated $m$ times to generate structured descriptions for a single image? What is the significance of this design? Alternatively, why is a dedicated MLP set up for each structured description?

---

> ### Author Response · Authors · 2025-11-21
>
> ## Embedding Fmri into a Text Space
>
> We agree that the NSD-general ROI primarily contains visual cortex regions, and the common belief is that visual stimuli should align better with vision-model embeddings. However, our argument is that while these regions process visual input, the internal representation of such input, especially as captured by fMRI, is not necessarily best aligned with vision model embeddings. This is likely because the internal representations formed by the human visual system may diverge systematically from how vision models encode images. Our empirical results show that fMRI signals from these regions align more closely with language model text space than with vision-based or joint vision-language spaces, which is the insight we want to convey through this paper. We hypothesize that the reason is that the border of the visual cortex acts as a convergence zone where information from the modal visual semantic system enters the amodal semantic system along a set of parallel, semantically selective pathways [1, 2]. Nonetheless, this interpretation needs further validation from the neuroscience community.
>
> ## More baselines and evaluation metrics
>
> We add two more baselines: 1. NeuralDiffuser(2025)[3], where a guidance strategy is developed for reconstruction, 2. MindBridge(2024) [4], where a biological-inspired aggregation function and a cyclic fMRI reconstruction mechanism is incorporated for reconstruction. The results for three datasets are shown in Tables R1, R2, R3, respectively. More evaluation metrics on NSD, including EfficientNet, SwAV, and Alex, are reported in Table R4.
>
>
> Table R1
> | **NSD** | PixCorr ↑ | SSIM ↑ | LPIPS ↓ | CLIP ↑ | Inception V3 ↑ |
> |--------|-----------|---------|----------|---------|----------------|
> | **PRISM** | 0.3404 +- 0.05 | 0.4640 +- 0.02 | 0.5963 +- 0.02 | 0.9467 +- 0.03 | 0.9516 +- 0.03 |
> | **MindBridge(2024)** | 0.1802 +- 0.03 | 0.2823 +- 0.02 | 0.6977 +- 0.03 | 0.9427 +- 0.02 | 0.9242 +- 0.03 |
> | **NeuralDiffuser(2025)** | 0.3011 +- 0.05 | 0.3348 +- 0.03 | 0.6522 +- 0.04 | 0.9409 +- 0.01 | 0.9487 +- 0.02 |
>
> Table R2
> | **BOLD5000** | PixCorr ↑ | SSIM ↑ | LPIPS ↓ | CLIP ↑ | Inception V3 ↑ |
> |--------------|-----------|---------|----------|---------|----------------|
> | **PRISM** | 0.2315 +- 0.01 | 0.5341 +- 0.02 | 0.6198 +- 0.02 | 0.7720 +- 0.03 | 0.6601+- 0.07 |
> | **MindBridge(2024)** | 0.1522 +- 0.04 | 0.3005 +- 0.01 | 0.6535 +- 0.03 | 0.7431 +- 0.01 | 0.6200 +- 0.04 |
> | **NeuralDiffuser(2025)** | 0.2036 +- 0.04 | 0.4005 +- 0.02 | 0.6899 +- 0.01 | 0.7609 +- 0.01 | 0.6522 +- 0.03 |
>
> Table R3
> | **GOD** | PixCorr ↑ | SSIM ↑ | LPIPS ↓ | CLIP ↑ | Inception V3 ↑ |
> |---------|-----------|---------|----------|---------|----------------|
> | **PRISM** | 0.2571 ± 0.01 | 0.5200 ± 0.02 | 0.6213 ± 0.01 | 0.8567 ± 0.05 | 0.8428 ± 0.06 |
> | **MindBridge(2024)** | 0.1898 +- 0.04 | 0.4227 +- 0.02 | 0.6960 +- 0.04 | 0.8219 +- 0.01 | 0.7960 +- 0.01 |
> | **NeuralDiffuser(2025)** | 0.2006 +- 0.02 | 0.4253 +- 0.03 | 0.6836 +- 0.02 | 0.8501 +- 0.01 | 0.8372 +- 0.03 |
>
>
>
>
>
> Table R4
>
> |                      | EfficientNet ↓ | SwAV ↓         | Alex(2) ↑      | Alex(5) ↑      |
> |----------------------|----------------|----------------|----------------|----------------|
> | **PRISM**                | 0.6020 +- 0.03 | 0.3128 +- 0.03 | 0.9692 +- 0.03 | 0.9881 +- 0.03 |
> | **Takagi**               | 0.7022 +- 0.03 | 0.3900 +- 0.03 | 0.8440 +- 0.04 | 0.8491 +- 0.03 |
> | **Mindvis**              | 0.6698 +- 0.01 | 0.3787 +- 0.03 | 0.8722 +- 0.05 | 0.8933 +- 0.04 |
> | **Mindeye1**             | 0.6302 +- 0.04 | 0.3426 +- 0.02 | 0.9566 +- 0.04 | 0.9748 +- 0.04 |
> | **MindBridge(2024)**     | 0.7033 +- 0.03 | 0.4064 +- 0.04 | 0.8754 +- 0.03 | 0.9553 +- 0.02 |
> | **Mindeye2**             | 0.6276 +- 0.03 | 0.3390 +- 0.03 | 0.9550 +- 0.02 | 0.9722 +- 0.03 |
> | **NeuralDiffuser(2025)** | 0.6439 +- 0.01 | 0.3887 +- 0.05 | 0.9584 +- 0.03 | 0.9833 +- 0.04 |

---

> ### Author Response · Authors · 2025-11-21
>
> ## Clarification on Formula Four
> In our framework, we use separate MLP encoders to extract distinct representations for each object in the viewed image. This design enables the model to isolate object-specific information from the fMRI signal, e.g., for an image containing two objects, we produce two separate representations, each guiding the generation of one object’s description and predicted location in the structured output. In contrast, using a single MLP yields only a single holistic representation that entangles all image-related information, which degrades performance. As shown in Table R5, mapping fMRI signals through a single MLP and generating the entire structured description with the language model leads to lower reconstruction quality.
>
> Table R5
> | Model | PixCorr ↑ | SSIM ↑ | LPIPS ↓ | CLIP ↑ | Inception V3 ↑ | EfficientNet ↓ | SwAV ↓ | Alex(2) ↑ | Alex(5) ↑ |
> |-------|-----------|---------|----------|---------|----------------|-----------------|--------|-----------|-----------|
> | **PRISM** | 0.3404 +- 0.05 | 0.4640 +- 0.02 | 0.5963 +- 0.02 | 0.9467 +- 0.03 | 0.9516 +- 0.03 | 0.6020 +- 0.03 | 0.3128 +- 0.03 | 0.9692 +- 0.03 | 0.9881 +- 0.03 |
> | **One MLP** | 0.3241 +- 0.03 | 0.4066 +- 0.02 | 0.6509 +- 0.01 | 0.9226 +- 0.05 | 0.9200 +- 0.04 | 0.6441 +- 0.04 | 0.3334 +- 0.02 | 0.9385 +- 0.04 | 0.9754 +- 0.02 |
>
> ## Case Study of Different Object Numbers
> We conduct case studies on different numbers of objects, and present the results in Appendix F.3. We observe that for more complex images (i.e., those containing four objects), setting m=2 encourages the VLM to describe the first object as a group of instances from the same category (e.g., “three players”), rather than generating separate descriptions for each individual object. In contrast, setting m=4 prompts the VLM to produce separate descriptions for each object, which in turn leads the generative model to synthesize a distinct latent representation for each. These representations are then concatenated according to their spatial locations and fed into the pre-trained diffusion model. However, when too many object-level features are introduced, the diffusion model tends to omit objects during the denoising process \cite{liu2024correcting, wang2023imagen}, resulting in incomplete reconstructions—for example, with one of three players missing in the final output.
>
>
> ## Clarification on Keyword Search
>
> To clarify the effectiveness of our keyword optimization strategy, we conducted an additional experiment in which the initial keyword set differed from that used in the main paper. Specifically, we excluded all keywords related to “spatial” in the initial set. The result is shown in Table R6; we can observe that the keyword search method still successfully discovers the keyword “spatial”. While the final keywords differ slightly from those in the main paper, they remain closely related to spatial relationships. This result highlights the robustness and effectiveness of our optimization strategy.
>
> R6
> | Rank | Round 0                 | Round 10               | Round 20                | Round 30               |
> |------|-------------------------|------------------------|-------------------------|------------------------|
> | #1   | Textural Attribute      | Spatial Relationship   | Spatial Arrangement     | Spatial Arrangement    |
> | #2   | Conceptual Linkage      | Textural Attribute     | Proximity Dynamics      | Proximity Relation     |
> | #3   | Action Interaction      | Visual Harmony         | Spatial Relationship    | Proximity Dynamics     |
> | #4   | Chromatic Feature       | Emotional Resonance    | Color Arrangement       | Spatial Relationship   |
> | #5   | Attribute Description   | Conceptual Linkage     | Hue Contrast            | Color Arrangement      |
>
>
>
>
> ## References
>
>
> [1] Popham, Sara F., et al. "Visual and linguistic semantic representations are aligned at the border of human visual cortex." Nature neuroscience
>
> [2] Tang, Jerry, et al. "Semantic reconstruction of continuous language from non-invasive brain recordings." Nature Neuroscience
>
> [3] Li, Haoyu, Hao Wu, and Badong Chen. "NeuralDiffuser: Neuroscience-Inspired Diffusion Guidance for fMRI Visual Reconstruction." IEEE Transactions on Image Processing. 2025.
>
> [4] Wang, Shizun, et al. "Mindbridge: A cross-subject brain decoding framework." Proceedings of the IEEE/CVF Conference on Computer Vision and Pattern Recognition. 2024.
>
> [5] Liu, Yujian, et al. "Correcting diffusion generation through resampling." Proceedings of the IEEE/CVF Conference on Computer Vision and Pattern Recognition
>
> [6] Wang, Su, et al. "Imagen editor and editbench: Advancing and evaluating text-guided image inpainting." Proceedings of the IEEE/CVF conference on computer vision and pattern recognition

---

> ### Comment · Reviewer_jrM3 · 2025-11-26
>
> Thanks very much for the author’s response and the additional results.
>
> The reply has addressed the vast majority of my concerns, especially those regarding the evaluation process. And I hope the authors can include these additional results in the final version.
>
> Overall, this paper innovatively introduces an object-centric generative model, enabling more precise fMRI-to-image reconstruction. The methodology is sound, and the evaluation is thorough. Based on this, I have raised the soundness and contribution scores to 3 and recommend that the paper be accepted by ICLR (I have raised my rating from 4 to 8).
>
> In addition, I suggest that the authors expand and discuss in more detail the recent papers on fMRI visual reconstruction that have been accepted by recent AI conferences (e.g., [1-5]) in the related work section, so as to provide a more comprehensive overview for future research.
>
> [1] Bao et al. Wills Aligner: Multi-Subject Collaborative Brain Visual Decoding. AAAI 2025.
>
> [2] Gong et al. MindTuner: Cross-Subject Visual Decoding with Visual Fingerprint and Semantic Correction. AAAI 2025.
>
> [3] Yeung et al. Reanimating Images using Neural Representations of Dynamic Stimuli. CVPR 2025.
>
> [4] Gong et al. NeuroClips: Towards high-fidelity and smooth fMRI-to-video reconstruction. NIPS 2024.
>
> [5] Quan et al. Psychometry: An omnifit model for image reconstruction from human brain activity. CVPR 2024.

---

> > ### Author Response · Authors · 2025-11-27
> >
> > Thank you very much for raising the score and for the helpful suggestions. We have cited the recommended paper and expanded the discussion accordingly; these changes are reflected in the revised Related Work section. We will also include the additional results in the final version. We appreciate your careful evaluation and constructive feedback.
> >
> >
> > Best,
> >
> > Authors of paper 15559

---

### Official Review · Reviewer_FGs4 · 2025-10-29

**Soundness:** 3
**Presentation:** 3
**Contribution:** 2
**Rating:** 8
**Confidence:** 3

**Summary:**

This paper introduces PRISM, a new framework for reconstructing visual images from fMRI signals. The authors find that fMRI data aligns more closely with structured text representations than with vision-only or joint vision-language spaces. The proposed method first decodes fMRI signals into an object-centric text description, which details objects, their attributes, and their relationships. This structured text then guides a specialized generative model to create the image, a process designed to improve the compositional accuracy of the reconstruction. The paper reports that this approach outperforms existing methods on three standard fMRI datasets.

**Strengths:**

The strengths include proposing a novel approach that challenges the assumption of using vision-based representations, instead providing evidence for the effectiveness of a structured text space for fMRI decoding. The method introduces specific components, such as an object-centric generative module and an attribute-relationship search, to directly address the known problem of compositional accuracy in image reconstruction. The paper's claims are supported by comprehensive quantitative evaluations across three different public datasets, showing improved performance over existing baseline methods, and are further validated by targeted ablation studies.

**Weaknesses:**

- The paper's central claim that fMRI signals align better with "text space" than "vision space"  rests on a specific comparison. The "text space" is generated using embeddings from image captions, which are already highly semantic, human-generated descriptions of the images. This is contrasted with the latent spaces of vision models (e.g., LDM, ResNet50). Consequently, the finding may be more precisely interpreted as fMRI signals aligning well with semantic, object-centric descriptions of images, rather than with language or text in a more general sense.

- The ablation study in Table 5 evaluates the impact of the "Attribute-Relationship Search" module. While the full PRISM model outperforms the "w/o AttOpt.+Bst" variant (which uses the best-performing initial keyword) , the performance difference between these two is smaller than the difference between the "best initial" and "worst initial" keywords. This suggests that while the search module adds value, a significant portion of the performance might be achievable by simply selecting a good, spatially-focused keyword from the initial set.

**Questions:**

- The paper demonstrates that fMRI signals align better with the T5 language model's text space than with the tested vision or joint-latent spaces. How dependent is this key finding on the specific choice of models? For instance, would this alignment hold if compared against different, perhaps more recent or semantically robust, vision-only representations?

- The entire training process for the fMRI-to-text encoder relies on structured descriptions generated by a pre-trained Vision-Language Model (VLM). How do errors, omissions, or inherent biases in this VLM's descriptions (which serve as the "ground truth" text) propag

- The object-centric diffusion module is designed to improve compositional accuracy. How does this method scale when reconstructing highly complex scenes containing many objects, significant overlap, or abstract relationships that are inherently difficult to capture in the structured text format?

---

> ### Author Response · Authors · 2025-11-21
>
> ## Clarification on Text Space
> While the text space indeed carries rich semantic information, our results indicate that its advantage cannot be attributed to semantics alone. This is evidenced by our comparison with vision–language models such as CLIP: although their representations also encode semantic information, they nonetheless show consistently weaker alignment with fMRI signals than pure text representations (Table 1). For this reason, we refer to this representational space as the “text space” rather than the “semantic space.”
>
> ## Alignment on Vision-only Representations
>
>
> Thanks for the suggestion, we include a more recent vision-only model Stable Diffusion 3 (SD3) [1], published in 2024. Alignment results are shown in Table R1, and reconstruction performance across latent spaces is reported in Table R2. As shown in Tables R1, R2, and Tables 1 and 4 in the paper, SD3 exhibits stronger alignment with fMRI signals compared to earlier vision models. However, there remains a consistent gap between SD3 and the text space of language models, which continue to show the highest alignment across all metrics. The alignment results for latent spaces can be seen in Table 1 and 4 of our paper.
>
>
> Table R1
> |            | CKA ↑   | Generalization gap ↓| CCA ↑   |
> |------------|--------|--------------------|--------|
> | **SD3**  | 0.2010 | 1.199             | 0.7289 |
> | T5 text (Used by ours)    | 0.5580  | 0.1132             | 0.8344 |
>
>
> Table R2
> |                        | PixCorr  ↑      | SSIM   ↑        | Lpips ↓          | CLIP ↑          | Inception V3 ↑  |
> |------------------------|----------------|----------------|----------------|----------------|----------------|
> | PRISM (ours)               | 0.3404 +- 0.05 | 0.464 +- 0.02  | 0.5943 +- 0.02 | 0.9467 +- 0.03 | 0.9516 +- 0.03 |
> | **SD3**                 | 0.2563   +- 0.05       | 0.4023  +- 0.05       | 0.7233 +- 0.02      | 0.8778 +- 0.04           | 0.9089 +- 0.01        |
>
>
> ## Errors in VLM
> Thank you for raising this point. The VLM used in our pipeline is required to have basic capabilities in image understanding and instruction following. However, once this condition is met, the quality of the generated structured descriptions depends less on the strength of the VLM itself and more on the specific object attributes and relationships used to guide the description process.
>
>
>
>
> ## Keyword Search
>
>
> To further demonstrate the effectiveness of our Attribute–Relationship Search module, we conducted an additional experiment where we excluded spatial-related keywords from the initial keyword set. We then evaluated whether our search procedure could still find spatially relevant terms. As shown in Table R3, the module successfully identifies spatial keywords even under this constraint. These results indicate that although including a strong spatially focused keyword in the initial pool can yield competitive performance, our method can automatically identify such effective keywords even when they are absent, demonstrating the robustness and effectiveness of our search process.
>
>
>
> Table R3
> | Rank | Round 0                 | Round 10               | Round 20                | Round 30               |
> |------|--|------|--------|-------|
> | #1   | Textural Attribute      | Spatial Relationship   | Spatial Arrangement     | Spatial Arrangement    |
> | #2   | Conceptual Linkage      | Textural Attribute     | Proximity Dynamics      | Proximity Relation     |
> | #3   | Action Interaction      | Visual Harmony         | Spatial Relationship    | Proximity Dynamics     |
> | #4   | Chromatic Feature       | Emotional Resonance    | Color Arrangement       | Spatial Relationship   |
> | #5   | Attribute Description   | Conceptual Linkage     | Hue Contrast            | Color Arrangement      |
>
>
>
> ## Case Study of Different Object Numbers
> We conduct case studies on different numbers of objects, and present the results in Appendix F.3. We observe that for more complex images (i.e., those containing four objects), setting m=2 encourages the VLM to describe the first object as a group of instances from the same category (e.g., “three players”), rather than generating separate descriptions for each individual object. In contrast, setting m=4 prompts the VLM to produce separate descriptions for each object, which in turn leads the generative model to synthesize a distinct latent representation for each. These representations are then concatenated according to their spatial locations and fed into the pre-trained diffusion model. However, when too many object-level features are introduced, the diffusion model tends to omit objects during the denoising process \cite{liu2024correcting, wang2023imagen}, resulting in incomplete reconstructions—for example, with one of three players missing in the final output.
>
>
> ## References
> [1] Esser, Patrick, et al. "Scaling rectified flow transformers for high-resolution image synthesis." Forty-first international conference on machine learning.

---

> > ### Comment · Reviewer_FGs4 · 2025-11-27
> >
> > Thanks for the reply. I will keep my rating.

---

### Official Review · Reviewer_M3K1 · 2025-10-31

**Soundness:** 3
**Presentation:** 3
**Contribution:** 3
**Rating:** 6
**Confidence:** 5

**Summary:**

The reconstruction of visual stimuli (images) from functional Magnetic Resonance Imaging (fMRI) signals involves a two-stage pipeline: transforming fMRI signals into a latent space and then using a pre-trained generative model to create the final image. The quality of the final reconstruction is largely determined by the alignment between the latent space and the structure of the neural activity, as well as the generative model's efficiency in producing images from that space. The proposed work demonstrates that accurate visual stimuli reconstruction can be achieved using the text space of a Large Language Model (LLM) instead of image-based latent representations. Furthermore, the reconstruction quality is further improved by object-centric diffusion that generates images by composing individual objects and identifying object attributes and relationships within the neural activity captured by the fMRI signals. Experiments on real-world datasets demonstrated that the proposed PRISM framework outperformed existing methods in CLIP metric by (in absolute terms) by 0-4% and in Inception V3 metric by 1-3%.

**Strengths:**

1. The paper makes an interesting observation that fMRI signals exhibit greater similarity to the text space of a language model than to either a vision-based space or a joint text-image space.

2. The pipeline for generating images that captures object relationships in the text space is intuitive.

3. The paper performs better than existing methods on many datasets, thus showing the importance of utilizing a structured text space as the an intermediate bridge between fMRI signals and image reconstruction.

**Weaknesses:**

1. The experimental results need to be clarified. How is the result for MindEye2+SDXL achieved? The original paper itself shows worse performance. The details regarding which NSD subject is used need to be provided.

2. Some other experimental comparisons need to be made:
[1] Wang, Shizun, et al. “Mindbridge: A cross-subject brain decoding framework.” Proceedings of the IEEE/CVF Conference on Computer Vision and Pattern Recognition. 2024.
[2] Gong, Zixuan, et al. “Mindtuner: Cross-subject visual decoding with visual fingerprint and semantic correction.” Proceedings of the AAAI Conference on Artificial Intelligence. Vol. 39. No. 13. 2025.

3. Results on other text spaces such as RoBERTa should be provided.

**Questions:**

Please refer to the weaknesses.

---

> ### Author Response · Authors · 2025-11-21
>
> ## Clarification on Reported Results for MindEye2+SDXL
> The higher performance reported for MindEye2+SDXL compared to the original paper is due to improvements we made to the original implementation. In particular, we incorporated a negative prompt in our model—a textual constraint that guides the diffusion model to avoid generating undesired visual artifacts (e.g., distorted object shapes or background clutter). For fairness, we applied the same negative prompt when evaluating MindEye2+SDXL, which led to the observed performance gains. When the negative prompt is removed, the reconstruction results (Table R1) closely match those reported in the original paper.
>
> Table R1: Performance of MindEye2+SDXL without negative prompts
> | **NSD** | PixCorr ↑ | SSIM ↑ | LPIPS ↓ | CLIP ↑ | Inception V3 ↑ |
> |--------|-----------|---------|----------|---------|----------------|
> | **MindEye2+SDXL** | 0.3311 +- 0.04 | 0.4211 +- 0.02 | 0.6133 +- 0.02 | 0.9289 +- 0.03 | 0.9500 +- 0.04 |
>
> ## More Baselines
> We include two more recent baselines: MindBridge (2024) [4], where a biological-inspired aggregation function and a cyclic fMRI reconstruction mechanism is incorporated for reconstruction, and NeuralDiffuser (2025) [3], where a guidance strategy is developed for reconstruction. Results on all three datasets are reported in Tables R2, R3, and R4, respectively. As the requested paper Mindtuner (Gong et al., 2025) [5] does not provide publicly available code, we instead include NeuralDiffuser, a method released around the same time, as the new baseline.
>
>
>
> Table R2
> | **NSD** | PixCorr ↑ | SSIM ↑ | LPIPS ↓ | CLIP ↑ | Inception V3 ↑ |
> |--------|-----------|---|--|---|---|
> | **PRISM** | 0.3404 +- 0.05 | 0.4640 +- 0.02 | 0.5963 +- 0.02 | 0.9467 +- 0.03 | 0.9516 +- 0.03 |
> | **MindBridge(2024)** | 0.1802 +- 0.03 | 0.2823 +- 0.02 | 0.6977 +- 0.03 | 0.9427 +- 0.02 | 0.9242 +- 0.03 |
> | **NeuralDiffuser(2025)** | 0.3011 +- 0.05 | 0.3348 +- 0.03 | 0.6522 +- 0.04 | 0.9409 +- 0.01 | 0.9487 +- 0.02 |
>
> Table R3
> | **BOLD5000** | PixCorr ↑ | SSIM ↑ | LPIPS ↓ | CLIP ↑ | Inception V3 ↑ |
> |--------------|-----------|---------|----------|---------|----------------|
> | **PRISM** | 0.2315 +- 0.01 | 0.5341 +- 0.02 | 0.6198 +- 0.02 | 0.7720 +- 0.03 | 0.6601 +- 0.07 |
> | **MindBridge(2024)** | 0.1522 +- 0.04 | 0.3005 +- 0.01 | 0.6535 +- 0.03 | 0.7431 +- 0.01 | 0.6200 +- 0.04 |
> | **NeuralDiffuser(2025)** | 0.2036 +- 0.04 | 0.4005 +- 0.02 | 0.6899 +- 0.01 | 0.7609 +- 0.01 | 0.6522 +- 0.03 |
>
> Table R4
> | **GOD** | PixCorr ↑ | SSIM ↑ | LPIPS ↓ | CLIP ↑ | Inception V3 ↑ |
> |---------|-----------|---------|----------|---------|----------------|
> | **PRISM** | 0.2571±0.01 | 0.5200±0.02 | 0.6213±0.01 | 0.8567±0.05 | 0.8428±0.06 |
> | **MindBridge(2024)** | 0.1898 +- 0.04 | 0.4227 +- 0.02 | 0.6960 +- 0.04 | 0.8219 +- 0.01 | 0.7960 +- 0.01 |
> | **NeuralDiffuser(2025)** | 0.2006 +- 0.02 | 0.4253 +- 0.03 | 0.6836 +- 0.02 | 0.8501 +- 0.01 | 0.8372 +- 0.03 |

---

> ### Author Response · Authors · 2025-11-21
>
> ## Results on Other Text Spaces
> We provide additional results on mapping input fMRI signals to the text spaces of RoBERTa [1] and Qwen3 [2]. Alignment results are shown in Table R5. Consistent with our main findings, the text space of language models exhibits the highest alignment with fMRI data, outperforming both vision-language and vision-only models across all metrics. We further conduct reconstruction experiments across these latent spaces, with results reported in Table R6. Note that since RoBERTa is not a generative model, it is excluded from this reconstruction comparison. Among the models evaluated, our proposed PRISM framework using T5 as the text space achieves the best reconstruction performance. Although Qwen3 achieves higher CKA and CCA scores, it exhibits a larger Generalization Gap than T5, which negatively impacts downstream reconstruction. In summary, these results reinforce our central finding: language model text spaces provide a more brain-aligned and effective intermediate representation for fMRI-to-image reconstruction.
>
> Table R5
> |            | CKA ↑   | Generalization Gap ↓| CCA ↑   |
> |------------|--------|--------------------|--------|
> | Qwen3 text  | 0.582  | 0.1204             | 0.8607 |
> | T5 text    | 0.558  | 0.1132             | 0.8344 |
> | Llama text | 0.5442 | 0.2216             | 0.8022 |
> | Roberta    | 0.53   | 0.2667             | 0.7988 |
> | Clip text  | 0.5177 | 0.4532             | 0.7599 |
> | Clip img   | 0.3668 | 0.486              | 0.7573 |
> | Vae        | 0.1957 | 1.252              | 0.7215 |
> | Resnet     | 0.1822 | 1.98               | 0.6746 |
>
>
> Table R6
> |                        | PixCorr ↑    | SSIM ↑           | Lpips ↓         | CLIP ↑          | Inception V3 ↑   |
> |----|----------------|----------------|----------------|----------------|----------------|
> | PRISM  | 0.3404 +- 0.05 | 0.464 +- 0.02  | 0.5943 +- 0.02 | 0.9467 +- 0.03 | 0.9516 +- 0.03 |
> | Qwen3| 0.332   +- 0.04       | 0.4023  +- 0.03       | 0.6317   +- 0.03      | 0.9300 +- 0.02           | 0.9299  +- 0.02       |
> |Clip text  | 0.3208 +- 0.04 | 0.3725 +- 0.06 | 0.6611 +- 0.05 | 0.9197 +- 0.02 | 0.9011 +- 0.04 |
> | LDM            | 0.2090 +- 0.07 | 0.3727 +- 0.07 | 0.7502 +- 0.04 | 0.8602 +- 0.06 | 0.8925 +- 0.05 |
>
>
> ## References
> [1] Yang, An, et al. "Qwen3 technical report."
>
> [2] Liu, Yinhan, et al. "Roberta: A robustly optimized bert pretraining approach."
>
> [3] Li, Haoyu, Hao Wu, and Badong Chen. "NeuralDiffuser: Neuroscience-Inspired Diffusion Guidance for fMRI Visual Reconstruction." IEEE Transactions on Image Processing. 2025
>
> [4] Wang, Shizun, et al. "Mindbridge: A cross-subject brain decoding framework." Proceedings of the IEEE/CVF Conference on Computer Vision and Pattern Recognition. 2024
>
> [5] Gong, Zixuan, et al. "Mindtuner: Cross-subject visual decoding with visual fingerprint and semantic correction." Proceedings of the AAAI Conference on Artificial Intelligence

---

> > ### Comment · Reviewer_M3K1 · 2025-11-27
> > **Thanks for the new experiments**
> >
> > I appreciate the results from the new experiments. I am raising my rating to 8.

---

### Meta-Review · Area_Chair_hUqK · 2026-01-06

**Summary:**

This submission tackles reconstruction of visual stimuli from fMRI data. It studies the importance of the latent semantic space used in this process, which should, among other things, match the brain representations.

The reviewers appreciated the insights from the paper on the representation methods as well as the empirical work which gives evidence to the claim, including the reconstruction performance of the method.

**Reviewer Concerns:**

The discussion was productive along many directions.

**Reviewer Scores:**

Two reviewers wanted to increase their rating from 6 to 8, another from 4 to 8. The resulting scores would have been [8, 8, 8, 8]

---

### Decision · Program_Chairs · 2026-01-26

Accept (Oral)